# MM-DeepResearch: A Simple and Effective Multimodal Agentic Search Baseline

Huanjin Yao [1]   Qixiang Yin [3]   Min Yang [1]   Ziwang Zhao [1]   Yibo Wang [4]   Haotian Luo [5]   Jingyi Zhang [4]   Jiaxing Huang [2]

## Abstract

We aim to develop a multimodal research agent capable of explicit reasoning and planning, multi-tool invocation, and cross-modal information synthesis, enabling it to conduct deep research tasks. However, we observe three main challenges in developing such agents: (1) scarcity of search-intensive multimodal QA data, (2) lack of effective search trajectories, and (3) prohibitive cost of training with online search APIs. To tackle them, we first propose **Hyper-Search**, a hypergraph-based QA generation method that models and connects visual and textual nodes within and across modalities, enabling to generate search-intensive multimodal QA pairs that require invoking various search tools to solve. Second, we introduce **DR-TTS**, which first decomposes search-involved tasks into several categories according to search tool types, and respectively optimize specialized search tool experts for each tool. It then recomposes tool experts to jointly explore search trajectories via tree search, producing trajectories that successfully solve complex tasks using various search tools. Third, we build an offline search engine supporting multiple search tools, enabling agentic reinforcement learning without using costly online search APIs. With the three designs, we develop **MM-DeepResearch**, a powerful multimodal deep research agent, and extensive results shows its superiority across benchmarks. Code is available at https://github.com/HJYao00/MM-DeepResearch.

---

[1]ByteDance  [2]Hong Kong Polytechnic University  [3]Zhongguancun Academy  [4]Nanyang Technological University  [5]Sichuan University. Correspondence to: Min Yang <yangminbupt@outlook.com>, Jiaxing Huang <jiaxing.huang@polyu.edu.hk>.

*Proceedings of the 43rd International Conference on Machine Learning*, Seoul, South Korea. PMLR 306, 2026. Copyright 2026 by the author(s).

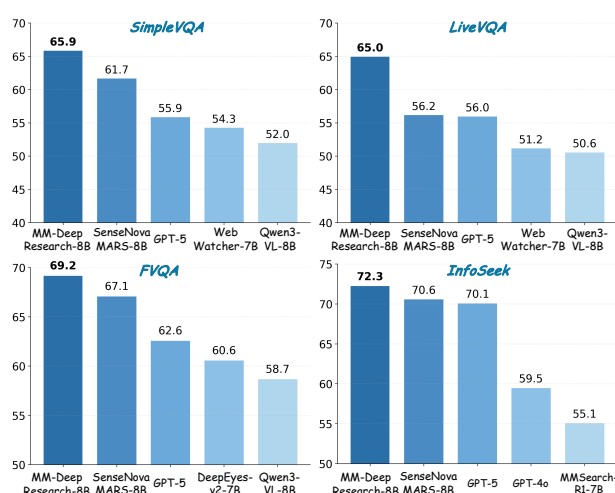

*Figure 1.* Overall performance of MM-DeepResearch-8B compares to other models across four benchmarks.

## 1. Introduction

Recently, reasoning Multimodal Large Language Models (MLLMs) (Jaech et al., 2024; Comanici et al., 2025) have shown remarkable capabilities in solving complex tasks by explicitly generating intermediate reasoning processes. Despite these advances, existing models remain fundamentally constrained by capacity-limited parameters with fixed and bounded knowledge, which restrict their ability to handle questions that go beyond their intrinsic knowledge, especially for information-intensive and open-world tasks.

To mitigate this limitation, early methods primarily adopted pre-defined information retrieval workflows, such as Retrieval-Augmented Generation (RAG) (Chen et al., 2024) or prompt-based approaches (Jiang et al., 2024), which retrieve external information as context in a pre-defined pipeline and then start reasoning upon the additional context. However, these approaches decouple retrieval from reasoning processes, lacking the ability to iteratively adapt retrieval strategies as the model's reasoning state evolves , resulting in limited search capability and generalization.

We aim to develop a multimodal deep research model equipped with agentic search capabilities, including explicit reasoning and planning, multi-tool invocation, and cross-modal information synthesis, enabling it to conduct deep re-

search tasks. We observe several fundamental challenges in developing such agents: (1) *Scarcity of search-intensive multimodal QA data.* Publicly available search-intensive multimodal QA datasets with multi-turn search and multi-tool invocation are very limited, resulting in insufficient supervision to effectively incentivize agentic search capabilities in MLLMs. (2) *Lack of effective search trajectories.* Traditional prompt-based search trajectory synthesis approaches are primarily designed for single-turn search, making them inadequate for multi-turn scenarios that require multi-round interaction with various search tools during iterative reasoning. (3) *Prohibitive cost of training with online search APIs.* Most current methods rely on online search APIs (e.g., SerpAPI and Jina) during training, which can easily incur thousands of dollars per training run, substantially limiting extensive experiments and systematic exploration.

To address these challenges, we propose three complementary techniques: Hyper-Search, DR-TTS, and an offline multimodal search engine. First, we propose **Hyper-Search**, a hypergraph-based method that models and connects visual and textual nodes within and across modalities, enabling to generate search-intensive multimodal QA pairs that require invoking various search tools to solve. Specifically, Hyper-Search jointly models web visual and textual content as nodes, and captures the native relationships among these nodes via hyperedges, enabling a structured representation of complex multi-source and multimodal information dependencies. Based on this hypergraph, Hyper-Search constructs search-intensive multimodal QA pairs by involving two or more hypergraph nodes across one or more hyperedges, ensuring that each generated question inherently requires multi-turn and multi-type search tool invocation to solve.

Second, we introduce Decompose–Recompose Tool Tree Search (DR-TTS) to effectively synthesize search-intensive reasoning trajectories that utilize various search tools to complete tasks. DR-TTS first decomposes search-involved tasks into several categories according to search tool types, and respectively train specialized search tool experts via reinforcement learning, with each expert mastering a single tool. This decomposition design simplifies learning complexity of search tools and enhances per-tool proficiency. DR-TTS then recomposes the tool experts to jointly explore and identify valid search trajectories via tree search, producing high-quality search trajectories for SFT. Compared to directly generating trajectories with a single model, DR-TTS mitigates tool-use bias and enhances per-tool proficiency, thus achieving more balanced and better exploration within and across tools, and significantly increases trajectory exploration diversity and success rates.

Third, upon above two designs, we construct an offline search engine that supports both information-based and knowledge-based search to handle a wide range of com-plex real-world tasks. The information-based search tools retrieve visual and textual context from pre-constructed multimodal corpus to support search-intensive tasks, while the knowledge-based search tool accepts complex search queries and return knowledge-intensive information generated by models. Compared with online search engines, our offline search engine provides faster tool responses and reduces tool-interactive training time. More importantly, it circumvents the expensive online search APIs cost, which can amount to thousands of dollars per training run.

Building on these designs, we train a powerful multimodal deep research agent named MM-DeepResearch. Trained with the search-intensive QA data generated Hyper-Search, the search tool trajectories synthesized via DR-TTS, and the offline search engine, our MM-DeepResearch achieves state-of-the-art performance across multiple benchmarks when evaluated using online search APIs, outperforming previous agents trained with costly online search APIs.

The main contributions of this work are fourfold. First, we propose Hyper-Search that introduces hypergraph to model and connect nodes within and across modalities, enabling to generate high-quality search-intensive multimodal QA pairs. To the best of our knowledge, this is the first work that introduces the concept of hypergraph for search-intensive data synthesis. Second, we design DR-TTS, which decomposes search-involved tasks to enable training specialized search tool experts, and recomposes tool experts to jointly explore and synthesize high-quality search trajectories. Third, we build an offline search engine with multiple search tools, enabling agentic reinforcement learning without using costly online search APIs. Fourth, extensive experiments demonstrate the superiority of our approaches and models across various multimodal deep research tasks.

## 2. Related Work

### 2.1. Agentic MLLMs

Early efforts leverage large-scale multimodal pretraining and instruction tuning to build general-purpose MLLMs for vision-language understanding tasks (Liu et al., 2024; An et al., 2025), and adopt post-training techniques to incentivize long-chain reasoning capabilities for complex reasoning tasks (Huang et al., 2025; Zhang et al., 2025a). Recent work further emphasizes agentic behaviors (Yao et al., 2025c), enabling MLLMs to autonomously and iteratively select and invoke tools for more complex tasks. Recent agentic foundational MLLMs (Team et al., 2026b), such as Qwen3-VL (Bai et al., 2025), are equipped with native tool-use capabilities by including diverse tool-interaction trajectories in agentic pretraining. In this paper, we aim to develop a multimodal deep research agent with improved capabilities in search tool invocation and multimodal infor-

mation synthesis.

## 2.2. Workflow-based Search Agents

Workflow-based search agents employ predefined and static information-seeking pipelines to retrieve external knowledge for reasoning. These approaches can be broadly categorized into RAG-based and prompt-based methods. (1) RAG-based approaches (Wang et al., 2025b; Li et al., 2025d) augment MLLMs by retrieving external knowledge from databases using similarity-based retrieval and injecting the retrieved content into the model input. (2) Prompt-based methods (Zheng et al., 2025; Wu et al., 2025b) explicitly orchestrate search behaviors through hand-crafted workflows encoded in prompts, where tool usage patterns are predefined during inference. These methods improve factual grounding and performance on knowledge-intensive tasks by incorporating external knowledge. However, workflow-based search agents decouple search from reasoning and are constrained by fixed and rigid pipelines, making them prone to insufficient or excessive search and consequently limiting their generalization performance. These limitations motivate us to explore multimodal deep research agents that actively and iteratively invoke search tools in a tightly coupled reasoning–search manner.

## 2.3. Multimodal Deep Research Agents

Recent work has introduced deep research agents that equip LLMs with agentic search capabilities for iterative information seeking and evidence synthesis through interaction with external text search tools, represented by OpenAI DeepResearch (OpenAI, 2025b) and MiroThinker (Team et al., 2025b). Beyond text-only agents, several prior works (Yang et al., 2025; Liu et al., 2025a; Xiao et al., 2025) extend deep research paradigms to multimodal domains. MMSearch-R1 (Wu et al., 2025a) first employs end-to-end RL to equip MLLMs with the ability to use image and text search tools. WebWatcher (Geng et al., 2025) synthesizes search-intensive QA pairs and converts them into VQA data for training multimodal deep research agents. However, most existing works remain closed-source in terms of search-intensive VQA data and search tool trajectories and depend on online search APIs, which pose significant challenges for the development of multimodal deep research agents. To tackle these challenges, this paper proposes Hyper-Search for generating multimodal search-intensive QA data, DR-TTS for synthesizing search trajectories, and an offline search engine for RL. Together, these components enable us to train a deep research agent from scratch and achieve strong empirical performance.

## 3. MM-DeepResearch Data

In this section, we focus on the data-level challenges of multimodal deep research. We first present a hypergraph-based method (Hyper-Search) for search intensive QA construction in Sec. 3.1. Then, we propose DR-TTS to jointly explore and find valid search tool trajectories in Sec. 3.2.

## 3.1. Hyper-Search for QA Data Generation

As shown in Figure 2, the construction of multimodal search-intensive QA data via Hyper-Search consists of three stages: (1) Hypergraph construction, which models web images, webpage content, and their relationships; (2) Multimodal QA generation, where QA pairs are generated based on the constructed hypergraph; and (3) Data filtering, which aims to retain high-quality, search-intensive QA data.

### 3.1.1. SEARCH HYPERGRAPH CONSTRUCTION

**Search Node Definition.** In Hyper-Search, we model web information using a hypergraph composed of two fundamental node types, *i.e.*, image nodes $\mathcal{I}$ and text nodes $\mathcal{T}$. Each image node $i_{d,k} \in \mathcal{I}$ corresponds to an image collected from an online source, while each text node $t_{d,k} \in \mathcal{T}$ represents the full content of a webpage, where $d$ denotes the expansion depth and $k$ indexes the nodes generated at that depth. To address redundancy in web content, we employ an MLLM to generate captions for image nodes and summaries for text nodes, which serve as concise representations during multimodal QA generation. These cross-modal representations enable modality-specific node expansion and interconnection, supporting the synthesis of multimodal QA pairs grounded in diverse information sources.

**Node Expansion.** We design distinct expansion strategies for image nodes and text nodes to simulate web-based information discovery between web images and webpage content. (a) *Image node expansion.* For each image node $i_{d,k}$, we define two expansion operations that respectively introduce new text and image nodes. The first operation, image reverse search, queries the image to retrieve $K$ relevant webpage contents, yielding text nodes $\{t_{d+1,1}^{i_{d,k}}, \ldots, t_{d+1,K}^{i_{d,k}}\}$. The second operation, image visual search, returns $K$ visually similar images, producing image nodes $\{i_{d+1,1}^{i_{d,k}}, \ldots, i_{d+1,K}^{i_{d,k}}\}$. (b) *Text node expansion.* For each text node $t_{d,k}$, we likewise design two expansion operations to extend the hypergraph with both text and image nodes. For text node expansion, we use an MLLM to extract the top-$K$ informative webpage URLs from the native page content, each introducing a new text node corresponding to the retrieved webpage $\{t_{d+1,1}^{t_{d,k}}, \ldots, t_{d+1,K}^{t_{d,k}}\}$. For image node expansion, in a similar manner, the LLM identifies the top-$K$ relevant image links directly from the native webpage content, which are used to form new image nodes $\{i_{d+1,1}^{t_{d,k}}, \ldots, i_{d+1,K}^{t_{d,k}}\}$.

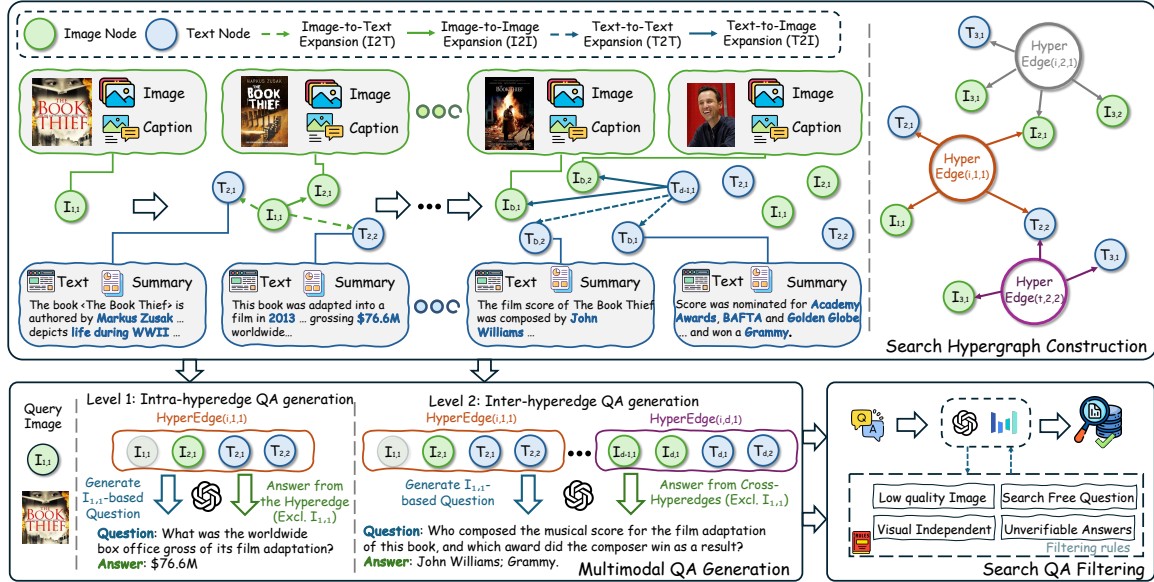

*Figure 2.* Overview of Hyper-Search for generating search-Intensive QA data via hypergraph construction, QA generation, and filtering. Here, $d$ denotes the current search depth, and $D$ denotes the maximum search depth.

**Hyperedge Connection.** For each node, we define a hyperedge $e_{d,k}$ that groups all nodes expanded from the same source via its expansion operations, capturing higher-order multimodal relationships. For example, given an image node $i_{d,k}$, the hyperedge is defined as $e_{d,k} = \{ i_{d,k}^{i_{d-1,k}}, i_{d+1,1}^{i_{d,k}}, \ldots, i_{d+1,K}^{i_{d,k}}, t_{d+1,1}^{i_{d,k}}, \ldots, t_{d+1,K}^{i_{d,k}} \}$. Hyperedges for text nodes are defined analogously.

**Summary.** Through continual expansion, Hyper-Search incrementally constructs a depth-$D$ hypergraph that organizes multimodal web information and provides structured relational units for downstream multimodal QA generation.

### 3.1.2. MULTIMODAL SEARCH QA GENERATION

QA generation is conducted at the hyperedge level, leveraging cross-modal and cross-source information within and across hyperedges to generate multimodal, search-intensive QA data. Based on the scope of aggregated evidence, *i.e.*, captions from image nodes and summaries from text nodes, we define two generation strategies: *intra-hyperedge* and *inter-hyperedge* QA generation.

**Level 1: Intra-hyperedge QA Generation.** At Level 1, QA generation is performed within a single hyperedge $e_{d,k}$. Specifically, we select an image node $i_{d,k}$ as the query image and generate QA data conditioned on the remaining image captions and webpage content within $e_{d,k}$. The generated search-intensive questions are explicitly vision-dependent on $i_{d,k}$, while the corresponding answers must be derived from the provided evidence. As the evidence span multiple modalities, this level of data effectively encourages the model to actively search using both image and text tools.

**Level 2: Inter-hyperedge QA generation.** At Level 2, we expand the scope of evidence sources to increase the search difficulty, and QA generation integrates evidence across multiple hyperedges $\{e_{d_1,k_1}, \ldots, e_{d_m,k_m}\}$. Specifically, an image node is selected as the query image, while the QA data is generated based on aggregated images and webpage contents drawn from these hyperedges. This level typically requires multiple rounds of search to resolve, further incentivizing agentic search capabilities that demand deeper exploration and more extensive information aggregation.

### 3.1.3. SEARCH QA FILTERING

To ensure data quality, we use MLLMs to filter low-quality multimodal QA instances. This process removes low-quality images, duplicated QA pairs, visually irrelevant or non-search-intensive questions, and answers unverifiable from the evidence. The resulting dataset contains 3K search-intensive multimodal QA pairs, termed **Hyper-Search-3K**.

### 3.2. DR-TTS for Search Trajectory Synthesis

We propose Decompose–Recompose Tool Tree Search (DR-TTS), a tree search method for synthesizing search tool trajectories inspired by the divide-and-conquer / mix of expert principle. DR-TTS first categorizes search-related tasks according to required search tools and decomposes model training to optimize specialized experts for each tool respectively. It then recomposes these tool experts to perform joint tree search, aiming to discover valid trajectories with higher exploration success rates and greater trajectory diversity.

**Search Tool Definition.** We define information-based and

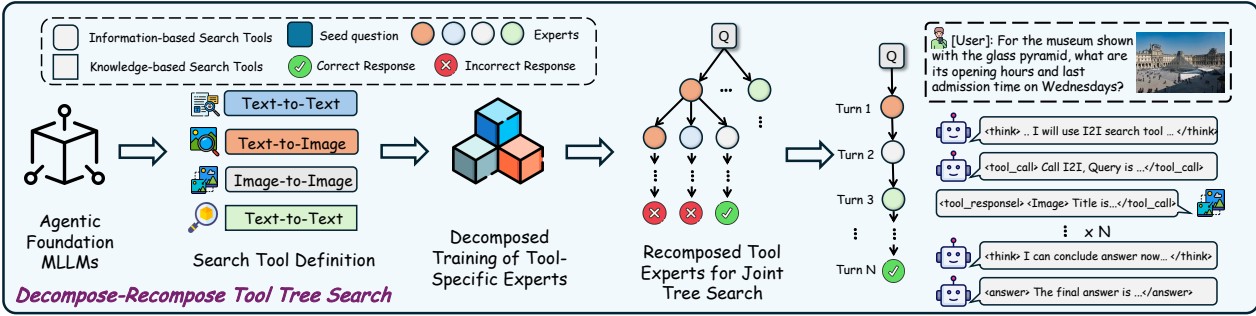

*Figure 3.* Overview of Decompose–Recompose Tool Tree Search for exploring search trajectories.

knowledge-based tools, which enable agents to acquire grounded factual information and specialized knowledge for more accurate trajectory search. *(1) Information-based search tools* retrieve real-world information by matching multimodal queries against external web sources. We design three types of information-based search tools (*i.e.*, text-to-text, text-to-image, and image-to-image retrieval) to obtain required textual and visual information based on different query types. For example, in text-to-image retrieval, agent generates a textual query to search for relevant web images. These tools provide grounded textual and visual evidence, enabling the model to address information-intensive reasoning tasks. *(2) Knowledge-based search tools* are designed to provide knowledge-intensive information by querying language models with complex queries. We formulate these tools as text-to-text interactions, in which our agents pose natural language queries and receive responses generated by domain-specific expert models. Such tools are particularly useful for domain-specific reasoning tasks that require information not directly accessible through web retrieval.

**Decomposed Training of Tool-Specific Experts.** Directly performing tree search with a pre-trained agentic foundational MLLM often leads to suboptimal performance, as the model frequently produces invalid tool calls and exhibits limited capability in integrating retrieved information, significantly hindering effective exploration. To address this issue, we first categorize search-related tasks according to the required search tools and decompose model training to optimize specialized experts for each tool. Specifically, we classify search-related training data by tool type using GPT. Each tool-specific subset is then used to train a corresponding expert via our RL strategy described in Sec. 4.2, without cold start. This process results in a set of $M$ specialized tool experts, each proficient in a single search tool, reducing learning complexity and enhancing per-tool proficiency.

**Recomposed Tool Experts for Joint Tree Search.** We then recompose these tool experts to collaboratively search for valid trajectories that solve the task. Starting from the root node $S_0 = [Q]$ (*i.e.*, the input question $Q = \{T, I\}$), we maintain a set of active nodes and expand them level by

**Algorithm 1** Search Trajectory Construction

**Input:** User Question $Q$, Tool Experts $\{E_1, E_2, \ldots, E_K\}$
**Output:** A valid search trajectory
1: Initialize root node $S_0 = [Q]$
2: Initialize active nodes $L = [S_0]$
3: **While** $L$ is not empty and max_depth not reached:
4:     new_nodes = [ ]
5:     **For each** node $S_t$ in $L$:
6:         **For each** expert $E_k$:
7:             thinking, action = $E_k(S_t)$
8:         **If** action is final_answer:
9:             **If** VerifyCorrectness(action) == True:
10:                 **Return** trajectory($S_0 \rightarrow \cdots \rightarrow S_{t+1}$)
11:         **Else**:
12:             response = Execute(action)
13:             **Create** $S_{t+1} = [$thinking, action, response$]$
14:             new_nodes.append($S_{t+1}$)
15:     $L$ = new_nodes

level. At each step, the shared context of a node is broadcast to all $M$ tool experts, and each expert proposes one next-step candidate based on its specialized tool. Therefore, each active node expands to at most $M$ child nodes.

We define the search tree as follows. The root node is the initial user question. Each intermediate node $S_t$ is a tuple [*thinking*, *tool_call*, *tool_response*], and each leaf node is a tuple [*thinking*, *final_answer*]. By recursively expanding nodes with expert-specific proposals, the search explores diverse expert-driven decisions and constructs candidate search trajectories. When any expert produces a final answer without invoking a tool, we verify its correctness using an LLM. If the answer is verified to be correct, the tree search terminates and returns the corresponding trajectory. Otherwise, that leaf node is treated as terminal and pruned from further expansion, while the search continues over the remaining active nodes.

**Search Trajectory Extraction.** For search trajectories that successfully produce a correct answer, we extract the corresponding explored path from the tree and construct trajec-

tory data for supervised fine-tuning (SFT). Each trajectory is represented as an ordered sequence of reasoning steps and tool interactions as $\tau = \{t_1, c_1, r_1, t_2, c_2, r_2, ..., t_n, a\}$. Through DR-TTS, tree-structured search over diverse tool experts enables the discovery of concise and effective reasoning trajectories by balancing tool usage during exploration. Ultimately, we collect 10K SFT trajectories using DR-TTS, referred to as DR-TTS-10K.

# 4. MM-DeepResearch Agents

Using the generated QA data and search tool trajectories, we train our MM-DeepResearch with a two-stage training recipe: (i) SFT on search tool trajectories as a cold start, and (ii) agentic multi-turn RL using our offline search engine.

## 4.1. SFT with Multi-turn Search Tool Invocation

Using search trajectories explored by DR-TTS, we perform multi-turn SFT as a cold start, training the model to learn tool call patterns and boost cross-modal information integration capabilities. The training objective maximizes likelihood of model-generated reasoning and planning steps $t_i$, search tool calls $c_i$, and the final answer $a_n$, while masking tool responses $r_i$ from the loss. The SFT objective is:

$$\mathcal{L}_{\text{SFT}} = -\mathbb{E}_{\tau \sim \mathcal{D}} \sum_{i=1}^{n} \log p_\theta(t_i, c_i, a_n \mid Q, t_{<i}, c_{<i}, r_{<i})$$
(1)

where the final answer $a_n$ is generated and supervised only at the final step, and tool responses $r_i$ are treated as observations conditioning the next-step prediction. In this way, the model learns appropriate tool-invocation patterns and improves long-context information synthesis.

## 4.2. Reinforcement Learning with offline Search Engine

Unlike prior methods that use online search APIs during RL, which can be prohibitively costly, especially for GRPO (Guo et al., 2025b) requiring multiple rollouts, we build an offline search engine to simulate real-world search environments and support both visual and textual retrieval.

**Offline Search Engine Construction.** To enable the offline search engine to invoke the tools defined in Sec. 3.2 during RL, we pre-collect a large-scale multimodal corpus comprising diverse images and texts. Specifically, for textual information, we use GPT to generate multiple candidate search queries and pre-fetch the corresponding webpage content. To further expand the corpus, we additionally incorporate Wikipedia data. For image-related information, we similarly pre-collect potentially relevant images and index them using a multimodal dense retrieval model. This corpus construction substantially reduces search API costs by avoiding repeated online API calls during training, while still support-

ing effective learning of agentic search and tool-invocation behaviors. Details of the offline corpus construction and retrieval implementation are provided in the appendix.

**Multi-Turn Reinforcement Learning.** To further incentivize model's deep research capabilities, we optimize it using multi-turn RL (*i.e.*, GRPO) with various tools. Specifically, given a question $Q$, the policy model generates a group of multi-turn search tool trajectories $\{\tau_1, \tau_2, \ldots, \tau_G\}$. These trajectories are executed concurrently via an asynchronous MLLM serving backend, while each trajectory proceeds synchronously in a turn-based manner, with the agent awaiting tool response at every step. Each trajectory consists of a sequence of reasoning and planning steps, tool invocations, tool responses, and a final answer.

**Reward Computation.** We then compute a reward $R_i$ for each trajectory $\tau_g$ that using a search-tool format reward $R_i^{format}$ and an outcome-level accuracy reward $R_i^{acc}$. (1) *Format reward.* The format reward encourages structured multi-turn reasoning and valid tool invocation by enforcing conformity to the expected interaction format. Trajectories that adhere to the reasoning format $\tau$ defined in Sec. 3.2 receive a reward of 1, whereas those that violate the format or produce unparsable tool-call queries receive a reward of 0. (2) *Accuracy reward.* The accuracy reward evaluates whether the final answer produced by the model is correct. Unlike previous methods that use rule-based evaluation, we extract the model's final answer enclosed within the <answer> and </answer> tags and assess its correctness using a powerful LLM by comparing it against the golden answer. The final reward is defined as $R = \alpha R_{acc} + (1 - \alpha)R_{format}$, where $\alpha$ balances the accuracy and format rewards. Subsequently, the group-relative advantage $A_g$ is computed according to the reward:

$$A_g = R_g - \frac{1}{G} \sum_{k=1}^{G} R_k.$$
(2)

**Optimization.** Finally, the GRPO employs a clipped objective with a KL penalty term to optimize the models:

$$\mathcal{J}_{\text{GRPO}}(\theta) = \mathbb{E}_{(I,T) \sim p_\mathcal{D}, \tau \sim \pi_{\theta_{\text{old}}}(\cdot | I, T)}$$

$$\left[ \frac{1}{G} \sum_{i=1}^{G} \min \left( \frac{\pi_\theta(\tau_i \mid I, T)}{\pi_{\theta_{\text{old}}}(\tau_i \mid I, T)} A_i, \text{clip} \left( \frac{\pi_\theta(\tau_i \mid I, T)}{\pi_{\theta_{\text{old}}}(\tau_i \mid I, T)}, \right. \right. \right.$$

$$\left. \left. \left. 1 - \epsilon, 1 + \epsilon \right) A_i - \beta D_{\text{KL}} \left( \pi_\theta || \pi_{\text{ref}} \right) \right) \right].$$
(3)

## 4.3. Evaluation Protocol

At test time, when using online search APIs, we employ an auxiliary LLM to verify and summarize tool responses before integrating them into the reasoning process, preventing excessively long outputs from exceeding the model's context length and causing reasoning failures.

*Table 1.* **Main Results.** To examine the effectiveness of our method, we compare our MM-DeepResearch with other deep research agents, as well as powerful MLLMs under direct reasoning and RAG worlflow paradigm. $^\dagger$ denotes results evaluated by ourselves.

| Method | SimpleVQA | MMSearch | LiveVQA | FVQA-test | InfoSeek | Browsecomp-VL | Average |
|---|---|---|---|---|---|---|---|
| *Direct Reasoning* | | | | | | | |
| GPT-4o (OpenAI, 2024) | 51.7 | 18.7 | 28.1 | 48.0 | 52.9 | 5.5 | 34.1 |
| GPT-5 (OpenAI, 2025a) | 54.1 | 35.1 | 44.4 | 54.4 | 61.7 | 48.6 | 49.7 |
| Qwen3-VL-8B (Bai et al., 2025) | 47.1 | 11.7 | 23.1 | 24.2 | 23.1 | 24.1 | 25.5 |
| Qwen3-VL-32B (Bai et al., 2025) | 58.0 | 19.8 | 45.5 | 34.1 | 28.0 | 30.8 | 36.0 |
| *RAG Workflow* | | | | | | | |
| GPT-4o (OpenAI, 2024) | 63.6 | 49.1 | 40.1 | 66.3 | 59.5 | 13.4 | 48.7 |
| GPT-5 (OpenAI, 2025a) | 55.9 | 52.6 | 56.0 | 62.6 | 70.6 | 54.9 | 58.8 |
| Qwen3-VL-8B (Bai et al., 2025) | 62.3 | 47.3 | 39.3 | 53.6 | 46.1 | 29.3 | 46.3 |
| *Agentic Search* | | | | | | | |
| DeepMMSearch-R1-7B (Narayan et al., 2025) | 55.8 | – | – | – | 47.5 | – | – |
| Visual-ARFT-7B (Liu et al., 2025b) | 42.4 | 34.5 | 25.4 | 41.7 | 37.9 | 16.5$^\dagger$ | 33.1 |
| MMSearch-R1-7B (Wu et al., 2025a) | 57.4 | 53.8 | 48.4 | 58.4 | 55.1 | 20.9$^\dagger$ | 49.0 |
| DeepEyes-v2-7B (Hong et al., 2025) | 59.4 | 63.7 | – | 60.6 | 51.1 | – | – |
| WebWatcher-7B (Geng et al., 2025) | 54.3 | 49.1 | 51.2 | – | – | 21.2 | – |
| MM-DeepResearch-7B | 62.0 | 61.4 | 60.0 | 61.9 | 58.7 | 32.8 | 56.1 |
| Qwen3-VL-8B (Bai et al., 2025) | 52.0 | 37.4 | 50.6 | 58.7 | 50.3 | 27.9 | 46.2 |
| SenseNova-MARS-8B (Chng et al., 2025) | 61.7 | 67.4 | 56.2 | 67.1 | 70.1 | 35.1$^\dagger$ | 59.6 |
| MM-DeepResearch-8B | 65.9 | 67.8 | 65.0 | 69.2 | 72.3 | 37.9 | 63.0 |
| Qwen3-VL-32B (Bai et al., 2025) | 58.7 | 44.4 | 45.5 | 60.2 | 58.5 | 35.1 | 50.4 |
| WebWatcher-32B (Geng et al., 2025) | 59.0 | 55.3 | 58.7 | – | – | 27.0 | – |
| MM-DeepResearch 32B | **67.6** | **69.0** | **68.0** | **70.1** | **73.9** | **43.0** | **65.3** |

## 5. Experiment

### 5.1. Dataset.

**The Source of Hyper-Search 3K.** To construct a comprehensive QA dataset, we collect visual sources from 7 diverse categories (*i.e.*, arts, sports, education, history, movies, places, and technology), as starting image nodes for Hyper-Search, ensuring broad domain coverage.

**Training Dataset.** We prepare training data for SFT and RL stages as follows. (1) For SFT, we select InfoSeek (Chen et al., 2023) for generating search tool trajectories. Before tree search, we select QA pairs that are challenging for Qwen3-VL-32B, and ultimately synthesize 10K valid search trajectories by DR-TTS for SFT. (2) For RL, we use 3K samples from Hyper-Search and an additional 3K search-intensive samples selected from FVQA (Wu et al., 2025a).

**Evaluation Dataset.** We evaluate and compare MM-DeepResearch on six information-intensive benchmarks that require invoking search tools to solve. Detailed dataset descriptions are provided in the Appendix.

### 5.2. Implementation Details

**Search Engine.** We detail the implementation of both online and offline search tools in this work. (1) *Online search.* We employ SerpAPI as the online search engine for Hyper-Search and DR-TTS to retrieve both textual and visual information from the web, owing to its robust and versatile search functionality. For image retrieval, we directly download the corresponding image files from the provided image URLs. For textual retrieval, since SerpAPI returns a list of web URLs, we further use Jina Reader to fetch and parse the content of the associated web pages. *(2) Offline Search.* We pre-collect multimodal web information and build an offline search engine that supports both text-based and multimodal retrieval. Text-based retrieval is performed using a text-only E5 embedding model (Wang et al., 2022), while multimodal retrieval is enabled by Jina-CLIP embeddings (Günther et al., 2025). The retrieval index is constructed using FlashRAG (Jin et al., 2025b).

**Base Model.** To examine the effectiveness of our search-intensive QA data, search trajectories, and offline search engine–based training, we select two kinds of MLLMs: (1) MLLMs without native agentic tool-use capabilities (*i.e.*, Qwen2.5-VL-7B-Instruct), which are used to assess whether our method can equip models with agentic search capabilities from scratch; (2) MLLMs with native agentic tool-use capabilities (*i.e.*, Qwen3-VL-8B and 32B-Instruct), which are used to examine whether our approach yields further performance gains beyond existing baselines.

**Training setup.** For SFT, we use LLaMA-Factory (Zheng et al., 2024) to fully fine-tune the model using a batch size of 128, a learning rate of 5e-6, and training over 3 epochs. For RL, we use VeRL (Sheng et al., 2024) with a global batch size of 128, 5 rollouts per prompt, and a learning rate of 1e-6. The maximum number of tool invocations is capped at 5 per

*Table 2.* **Ablation Study on Hyper-Search.**

| Method | Avg. Tool Calls | MMSearch |
|---|---|---|
| InfoSeek | 1.6 | 62.1 |
| Graph-based | 1.7 | 64.8 |
| Hyper-Search | 2.3 | 67.8 |

*Table 3.* **Ablation study on MM-DeepResearch.**

| Method | DR-TTS (SFT) | Hyper-Search (RL) | MMSearch |
|---|---|---|---|
| Qwen3-VL (Baseline) | – | – | 37.4 |
| MM-DeepResearch-SFT | ✔ | – | 52.3 |
| MM-DeepResearch-RL | – | ✔ | 65.2 |
| MM-DeepResearch | ✔ | ✔ | 67.8 |

*Table 4.* **Ablation study of training cost and efficiency for online and offline search.**

| Method | Training Cost ($) | Training Time (s / sample) |
|---|---|---|
| Online Search | 640 | 60 |
| Offline Search | 0 | 1 |

*Table 5.* **Ablation Study of offline search tools.**

| Information-based | | | Knowledge-based | MMSearch |
|---|---|---|---|---|
| T2T | T2I | I2T | T2T | |
| – | – | – | – | 11.7 |
| ✔ | – | – | – | 55.9 |
| ✔ | ✔ | – | – | 64.7 |
| ✔ | ✔ | ✔ | – | 66.9 |
| ✔ | ✔ | ✔ | ✔ | 67.8 |

trajectory. To handle long multimodal tool responses and enhance long-context information integration capabilities, we scale the maximum training context length to 70,000 tokens. For smaller MLLMs (*e.g.*, 7B and 8B), both SFT and RL are conducted on 8 NVIDIA H100 GPUs, while larger models (*e.g.*, 32B) are trained on 32 NVIDIA H100 GPUs. In addition, we allocate 8 NVIDIA H100 GPUs to deploy the offline search engine and judge model.

## 5.3. Main Results

To evaluate our method, we conduct experiments on models without and with native agentic tool-use capabilities, and compare our MM-DeepResearch against various SOTAs, including non-agentic and agentic search MLLMs in Tab. 1.

**Compared with non-agentic models.** We first conduct experiments on Qwen2.5-VL-7B, which lacks native tool-calling capabilities. Trained on our synthesized search trajectories and generated QA data, MM-DeepResearch-7B learns agentic search capabilities, enabling multi-step reasoning and the coordinated use of diverse tools to retrieve the required information and synthesize final answers, which significantly improves its performance on deep research tasks. Under the same base model, MM-DeepResearch-7B outperforms prior agentic search MLLMs, i.e., Visual-ARFT and MMSearch-R1-7B, with average improvements of 23% and 7.1% across six benchmarks. Moreover, it surpasses the previous 7B state-of-the-art WebWatcher by 7.7% and 12.3% on SimpleVQA and MM-Search, respectively. These results demonstrate the effectiveness of our approach in incentivizing agentic search capabilities from scratch.

**Compared with agentic models.** We further evaluate our approach on agentic foundational MLLMs (i.e., Qwen3-VL-8B and Qwen3-VL-32B), which possess basic tool-use capabilities, to assess whether our approach can further enhance their performance. As shown in Tab. 1, MM-DeepResearch-8B averagely improves upon baseline Qwen3-VL-8B by 17%. Compared with deep research agent SenseNova-MARS-8B, MM-DeepResearch-8B achieves an average gain of 3.4 points, with improvements of 4.2% on Sim-

pleVQA. In addition, when scaling the model to 32B, MM-DeepResearch-32B exhibits consistent performance gains of 14.9% over baseline Qwen3-VL-32B, further validating its effectiveness in enhancing agentic foundational MLLMs.

## 5.4. Ablation Study

**Ablation study on Hyper-Search.** We analyze how different data generation methods affect search depth (*i.e.*, average tool calls) and performance in Tab. 2. We compare RL training results using different datasets, *i.e.*, InfoSeek, graph-based, and Hyper-Search, each containing 3K samples and initialized from our SFT model. The results show that RL training on data constructed via Hyper-Search enables deeper exploratory behavior, encouraging more search turns and leading to more accurate answers grounded in more comprehensive information.

**Ablation study of MM-DeepResearch.** We study the contributions of different components in MM-DeepResearch in Tab. 3. Starting from the Qwen3-VL baseline (37.4 on MMSearch), training with DR-TTS-explored search trajectories for SFT substantially improves performance to 52.3, indicating that our search trajectory data effectively enhances search behavior and information integration. When trained only with Hyper-Search data using RL, MM-DeepResearch-RL further boosts performance to 65.2, demonstrating that the dataset incentivize multi-turn agentic search. By jointly leveraging DR-TTS trajectory data for SFT and Hyper-Search QA data for RL, MM-DeepResearch achieves the best performance, reaching 67.8 on MMSearch, validating the effectiveness of our proposed methods.

**Ablation study on Search Tools.** Tab. 5 presents an ablation study of offline search tools. Enabling information-based search progressively improves MMSearch performance, with T2T, T2I, and I2T increasing the score from 11.7 to 66.9. Adding knowledge-based T2T search further boosts performance to 67.8, demonstrating the comple-

mentary effects of information-based and knowledge-based search in multimodal reasoning.

**Ablation Study of Online and Offline Search Costs.** Tab. 4 compares training costs for online and offline search during RL. Online search incurs substantial costs (around 640 dollars per hundred steps) due to repeated external API calls in each rollout and also introduces high latency, with an average response time of about 60 seconds per sample. In contrast, offline search operates with no additional search-related cost and significantly lower latency (about 1 second per sample). These results show that offline search is a cost-efficient and scalable alternative, particularly for GRPO settings with multiple rollouts and large-scale training.

# 6. Conclusion

This paper presents MM-DeepResearch, an agent designed for deep research tasks that integrates reasoning and planning, multi-turn search tool invocation, and long-context cross-modal information integration. To develop MM-DeepResearch, we introduce three core components: Hyper-Search for search-intensive QA generation, DR-TTS for synthesizing effective search trajectories, and an offline search engine for scalable RL training. Together, these designs enable training a search agent from scratch and substantially enhance its search and reasoning capabilities. Extensive experiments demonstrate the effectiveness of our approach across models with and without native agentic capabilities. We hope this work provides a simple yet effective baseline for future research on multimodal deep research agents.

# Impact Statement

This paper presents work whose goal is to advance the field of Multimodal Large Language Models and Agentic AI. There are many potential societal consequences of our work, none which we feel must be specifically highlighted here.

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

# Appendix

## A. Multimodal QA Dataset

To construct a comprehensive QA dataset, we collect visual sources from seven diverse categories (*i.e.*, arts, sports, education, history, movies, places, and technology), as starting image nodes for Hyper-Search, ensuring broad domain coverage. We provide more examples extracted from our Multimodal QA Dataset. In Fig. 4 ∼ Fig. 10, we try to cover different QA categories in every task to offer a holistic overview of Multimodal QA Dataset.

### *History*

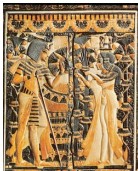 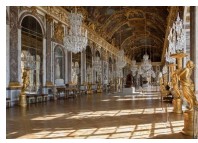 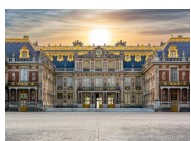 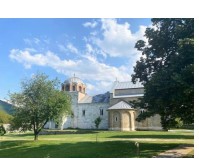

**Node1**
**Question:** If you plan to visit the landmark shown, which version of the Paris Pass includes entry to it, and how many total attractions does that version cover?
**Answer:** Nebkheperure

**Node2**
**Question:** For the mirrored gallery shown, who was its architect and principal decorator, and during which years was it constructed?
**Answer:** Architect: Jules Hardouin-Mansart; decorator: Charles Le Brun; built 1678–1684

**Node3**
**Question:** What is the name of the grand palace shown in this image with gold-trimmed roofs and a symmetrical central facade?
**Answer:** Palace of Versailles (Château de Versailles)

**Node4**
**Question:** What is the name of the monastery complex shown in the image?
**Answer:** Studenica Monastery

*Figure 4.* Data examples from the history category.

### *Art*

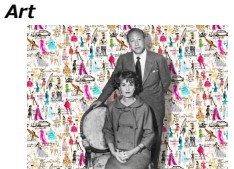 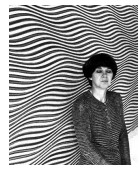 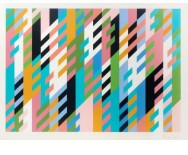 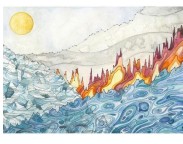

**Node1**
**Question:** What visual art technique is exemplified by the black-and-white couple's photograph superimposed on a colorful illustrated backdrop, and which movement first widely adopted it for political protest?
**Answer:** Photomontage; Dada artists around 1915–1916

**Node2**
**Question:** Observing the black-and-white wavy optical pattern in the background—characteristic of a British Op Art painter who pioneered such geometric illusions in the early 1960s—which artists' organization did she co-found in 1968?
**Answer:** SPACE

**Node3**
**Question:** Observing the multicolored geometric lithograph shown, in what year did its artist first shift from black-and-white works to working in color?
**Answer:** 1967

**Node4**
**Question:** What is the title of this watercolor artwork that visualizes climate change data as a landscape?
**Answer:** Landscape of Change

*Figure 5.* Data examples from the art category.

### *Education*

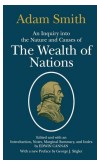 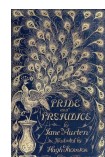 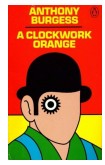 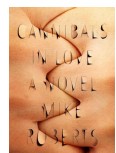

**Node1**
**Question:** Viewing this cover of The Wealth of Nations, according to the Porter Square Books article ranking book covers by Pinterest engagement, what position does it hold and what ISBN is listed for it?
**Answer:** #1; ISBN 9780679783367

**Node2**
**Question:** This ornate peacock-feather cover of Pride and Prejudice is from a historic edition—what year was this famous version first published?
**Answer:** 1894

**Node3**
**Question:** On the cover showing a bowler-hatted figure with a gear-like eye, what specific symbolic character element do design analyses cite as encapsulating the novel's core themes?
**Answer:** a cog-eyed droog

**Node4**
**Question:** The book cover shown is for Mike Roberts's novel "Cannibals in Love." Who published this edition, and on what release date did it come out?
**Answer:** FSG Originals (Macmillan); September 20, 2016

*Figure 6.* Data examples from the education category.

### Movie

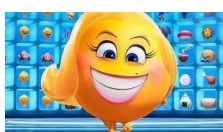 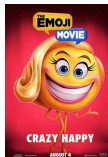 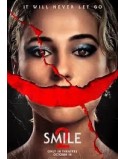 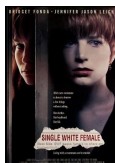

**Node1**
**Question:** What is the name of the blonde, wide-smiling emoji character shown here from the emoji-themed animated film?
**Answer:** Smiler (The Emoji Movie)

**Node2**
**Question:** The cheerful blonde emoji character shown in this poster—within The Emoji Movie, what is her role and who voices her?
**Answer:** Main antagonist; voiced by Maya Rudolph

**Node3**
**Question:** On the movie poster shown with the title "SMILE," what U.S. theatrical release date is announced on the official poster?
**Answer:** September 30

**Node4**
**Question:** The image shows a poster for the film depicted; during its production, which Manhattan building was used for exterior shots?
**Answer:** The Ansonia Hotel

*Figure 7.* Data examples from the movie category.

### Places

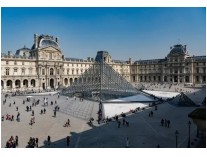 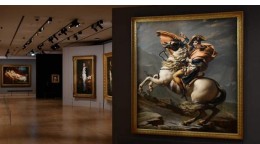 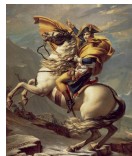 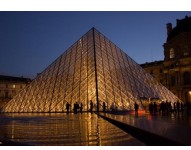

**Node1**
**Question:** For the museum shown with the glass pyramid, what are its opening hours and last admission time on Wednesdays?
**Answer:** Wednesday: 9:00–21:00; last entry 20:00

**Node2**
**Question:** In the Louvre gallery displaying Jacques-Louis David paintings seen here, which two major loaned works are featured in this exhibition?
**Answer:** The Tennis Court Oath (Versailles) and the original Death of Marat (Royal Museums of Fine Arts of Belgium, Brussels)

**Node3**
**Question:** In this equestrian painting of a general on a rearing white horse crossing the Alps, which historical leaders was the image intended to associate him with according to art historical analysis?
**Answer:** Hannibal and Charlemagne

**Node4**
**Question:** The illuminated glass pyramid shown here is the main entrance to this museum—who designed it, and under which expansion program was it built?
**Answer:** I.M. Pei; Grand Louvre project (1980s–1990s)

*Figure 8.* Data examples from the places category.

### Technology

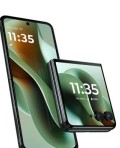 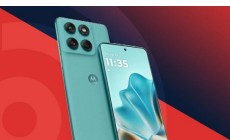 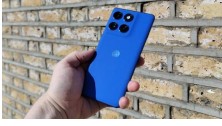 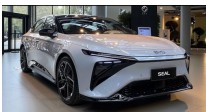

**Node1**
**Question:** For the foldable smartphone shown with both its large inner screen and small outer cover screen visible, what refresh rate does the cover display support?
**Answer:** 165Hz

**Node2**
**Question:** The teal Motorola phone shown has a 2x2 quad-camera module like the Edge 60 Pro; for shoppers in the United States where this model isn't sold, which two Motorola phones are recommended alternatives?
**Answer:** Motorola Razr Ultra 2025; Moto G75 5G

**Node3**
**Question:** The blue Motorola smartphone shown has a square camera module with four lenses; what official durability certifications does this model carry?
**Answer:** IP69 and MIL-STD-810H

**Node4**
**Question:** In the Philippine market, what is the official vehicle warranty term (years and total kilometers) for the 2026 rear-wheel-drive "Advance" trim of the electric sedan shown?
**Answer:** 6 years / 150,000 km

*Figure 9.* Data examples from the technology category.

### Sports

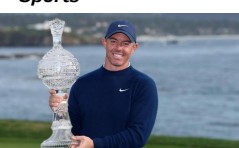 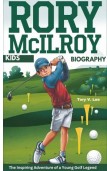 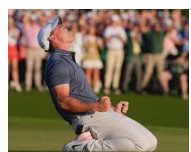 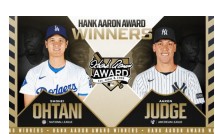

**Node1**
**Question:** The image shows a professional golfer holding a crystal trophy at a coastal course; according to his official schedule, when and where is he slated to play the AT&T Pebble Beach Pro-Am in 2026?
**Answer:** 12-15 February 2026, Pebble Beach, California, USA

**Node2**
**Question:** Looking at this book cover featuring a golfer's biography, what notable career milestone does that golfer achieve as highlighted in the PGA TOUR Studios video "The Long Game"?
**Answer:** Sixth man to complete the career Grand Slam

**Node3**
**Question:** For the golfer celebrating on the green, according to related visuals of the same golf attire, what brand logo is on the white golf glove typically worn?
**Answer:** Nike swoosh

**Node4**
**Question:** The image shows a promotional graphic announcing two MLB players as Hank Aaron Award winners; in what year were these winners recognized?
**Answer:** 2024

*Figure 10.* Data examples from the sports category.

## B. More Search Engine Setup

**Offline Search Engine Implementation.** To enable offline corpus search, we pre-collect a multimodal corpus and adopt a retrieval framework based on FlashRAG (Jin et al., 2025b). For the textual corpus, we prompt GPT to generate multiple candidate search queries for each question and use SerpAPI[1] and Jina[2] to retrieve the corresponding webpage content. In addition, we incorporate large-scale English Wikipedia (2018) to improve corpus coverage and diversity, simulating the large-scale information available on the web. These web contents are segmented into 512-word passages with titles appended and employ e5-base-v2 (Wang et al., 2022) as the text retriever. For the image corpus, we additionally prompt GPT to generate multiple candidate image search queries and use SerpAPI to retrieve relevant image URLs. We employ jina-embeddings-v4 (Günther et al., 2025) as the multimodal retriever to enable cross-modal retrieval.

**Hyperparameters of Search Engine.** During training, we retrieve the top-3 most similar textual passages for each query. For image retrieval, only images with an embedding similarity greater than 0.7 are retained, and for each tool call query we keep only the single most similar image to avoid excessively long reasoning contexts and excessive memory consumption. During evaluation, we adopt online retrieval, expanding textual search to the top-5 results and retrieving up to three images per query.

## C. Evaluation Benchmark

We evaluate MM-DeepResearch across six benchmarks, which are described as follows.

- **SimpleVQA.** SimpleVQA (Cheng et al., 2025) is a newly proposed multimodal benchmark designed to evaluate the factuality of MLLMs when answering short natural-language questions grounded in images. It encompasses diverse tasks and scenarios, with high quality and challenging queries aligned across multiple topics. The benchmark ensures static, verifiable reference answers and employs rigorous quality controls, making it straightforward to evaluate model outputs with minimal variance. Using a combination of manual verification and an LLM as a judge scoring system, SimpleVQA facilitates comprehensive empirical comparisons across leading MLLMs and text only LLMs, providing insight into image comprehension, generative accuracy, and common failure modes. Following WebWatcher (Geng et al., 2025), we evaluate on the same 300 examples sampled from the 1,013 English QA pairs.

- **MMSearch.** MMSearch (Jiang et al., 2024) represents a multimodal browsing benchmark crafted to place rigorous demands on models that must integrate visual and textual signals with external retrieval to answer complex queries. Unlike prior multimodal tasks where salient objects may be easily recognizable, this benchmark requires exhaustive reasoning over localized visual cues, disciplined cross modal evidence gathering, provenance verification under noisy search results, and long, tool augmented reasoning chains. Among them, 171 examples include images, and we evaluate on these 171 image-based examples to align with previous methods.

- **LiveVQA.** LiveVQA (Fu et al., 2025) is a large scale visual question answering dataset that focuses on up to date visual knowledge. It contains over 107,000 samples across multiple real world categories sourced from recent news, video, and academic platforms, making it a benchmark tailored for evaluating how MLLMs handle real time visual information beyond their training cutoffs. The dataset supports studies on both performance gaps in current models and parameter efficient fine tuning strategies that balance new visual knowledge integration with core perception and reasoning capabilities. We also evaluate on a 300-example subset sampled from WebWatcher.

- **FVQA.** FVQA (Wang et al., 2017) is a foundational visual question answering dataset that extends traditional VQA tasks by requiring external commonsense knowledge to answer visual questions correctly. In FVQA, each image–question–answer triplet is accompanied by a supporting fact represented as a structured knowledge triplet, drawn from common sense knowledge bases. This design forces models to reason beyond visual perception and to retrieve and integrate factual knowledge, thus serving as a core benchmark for knowledge based VQA research involving explicit external knowledge grounding. We test on the full set of 1,800 high-quality examples here.

- **InfoSeek.** InfoSeek (Chen et al., 2023) is a visual question answering dataset that concentrates on information seeking questions – queries that cannot be resolved using only common sense or visual content presented in the image. The dataset blends human annotated and large scale automatically generated question–answer pairs, challenging models

---

[1]SerpAPI: https://serpapi.com/

[2]Jina Reader: https://jina.ai/reader/

to leverage fine grained knowledge beyond typical VQA tasks. Evaluation using InfoSeek highlights the limitations of current pre trained vision language models in answering knowledge intensive questions, and underscores the performance gains enabled by fine tuning with explicit information seeking objectives. Here, we use 2,000 instances for evaluation, sampled by MMSearch-R1 (Wu et al., 2025a).

- **Browsecomp-VL.** BrowseComp-VL (Geng et al., 2025) is a recently introduced benchmark that extends BrowseComp style tasks into the multimodal domain, requiring strategic planning and cross modal retrieval across text, images, and web content. Designed to evaluate sophisticated multimodal agents, BrowseComp-VL demands deep search, sequential evidence synthesis, and multimodal reasoning. By embedding both visual and textual clues into long horizon information seeking problems, it assesses agents' ability to coordinate perception, retrieval, and planning in complex real world scenarios, and serves as an evaluation frontier for integrated vision language deep research. We evaluate on the full BrowseComp-VL dataset.

## D. More Discussion

### D.1. Discussion on Different Embedding Models for the Offline Search Engine

We investigate the impact of different embedding models for offline text retrieval, with results reported in Table 6. Overall, the performance differences between the two embedding models are marginal, indicating that offline text retrieval is relatively robust to the choice of embedding model. Although jina-embeddings-v4 (Günther et al., 2025) achieves slightly better performance, it incurs higher cost due to its high-dimensional embeddings and slower inference speed. Considering efficiency and scalability, we therefore adopt e5-base-v2 (Wang et al., 2022) for text embedding, which provides comparable retrieval quality while being significantly more efficient for large-scale offline indexing and retrieval.

*Table 6.* **Discussion on Different Embedding Models for the Offline Search Engine.**

| Embedding Model | MMSearch |
|---|---|
| e5-base-v2 | 55.9 |
| jina-embeddings-v4 | 56.1 |

### D.2. Comparing Online and Offline Search Performance at Evaluation

We compare the performance of online and offline search engines during evaluation, as shown in Table 7. Overall, online search consistently achieves better performance, benefiting from access to up-to-date and large-scale web information. Nevertheless, the offline search engine attains competitive results on both SimpleVQA and MMSearch, demonstrating its ability to retrieve informative and relevant evidence. These results indicate that although offline search does not fully match online search performance, it is sufficiently effective to provide meaningful retrieval signals and can adequately stimulate the model to learn search behaviors. This makes offline search a practical and cost-efficient alternative for training.

*Table 7.* **Discussion of Offline Text Search Engine.**

| Search Engine | SimpleVQA | MMSearch |
|---|---|---|
| Offline | 63.4 | 62.7 |
| Online | 65.9 | 67.8 |

### D.3. Discussion on the Impact of Top-$k$ Text Retrieval during Training

We analyze the effect of varying the number of retrieved textual passages ($k$) during training, with results shown in Table 8. Overall, moderate values of $k$ lead to better performance, while increasing $k$ beyond a certain point yields diminishing returns. Specifically, retrieving five passages achieves the best performance on MMSearch, outperforming both smaller ($k = 3$) and larger ($k = 10$) settings. When $k$ becomes too large, the inclusion of excessive and potentially noisy information may dilute relevant evidence and increase reasoning difficulty, leading to performance degradation.

*Table 8.* **Discussion of Offline Text Search Engine.**

| Embedding Model | Top-$k$ | MMSearch |
|---|---|---|
| e5 | 3 | 55.9 |
| e5 | 5 | 56.3 |
| e5 | 10 | 54.8 |

### D.4. Ablation Results on Additional Benchmarks

Table 9 reports a more detailed version of the ablation study in Table 3 by including additional benchmark results. Overall, the trends are fully consistent with those in Table 3. Both MM-DeepResearch-SFT and MM-DeepResearch-RL consistently outperform the base model Qwen3-VL-8B, while the full MM-DeepResearch model achieves the strongest performance across all reported benchmarks. These results provide additional evidence that supervised fine-tuning on search-intensive trajectories and reinforcement learning are complementary, and together produce the most robust gains.

*Table 9.* **Performance comparison on multimodal search benchmarks.**

| Method | SimpleVQA | MMSearch | LiveVQA | FVQA-test | InfoSeek | Browsecomp-VL |
|---|---|---|---|---|---|---|
| Qwen3-VL-8B | 52.0 | 37.4 | 50.6 | 58.7 | 50.3 | 27.9 |
| MM-DeepResearch-SFT | 59.2 | 52.3 | 52.7 | 59.4 | 61.8 | 31.3 |
| MM-DeepResearch-RL | 63.4 | 65.2 | 63.4 | 65.8 | 70.1 | 33.0 |
| **MM-DeepResearch** | **65.9** | **67.8** | **65.0** | **69.2** | **72.3** | **37.9** |

### D.5. Ablation on Individual Search Tools

We further conduct an ablation study on individual search tools by enabling each tool alone. Compared with the no-tool baseline, all tools improve performance, with text-to-text retrieval based on the information retrieval system bringing the largest gain, followed by image-to-image retrieval. Text-to-image and image-to-text retrieval are less effective individually, but still contribute positively. Notably, using all four tools together achieves the best overall result, clearly surpassing every single-tool setting. This suggests that the four tools provide complementary evidence from different retrieval channels, and their combination is essential for maximizing performance.

*Table 10.* **Ablation on individual search tools.**

| Metric | w/o Tool | T2T(info) | T2I | I2I | T2T(model) | 4 Tools |
|---|---|---|---|---|---|---|
| Acc | 11.7 | 55.9 | 26.8 | 38.1 | 32.7 | **67.8** |

## E. Case Study

We present the inference cases of MM-DeepResearch in Fig. 11. In the reasoning trajectory, MM-DeepResearch alternates between thinking and tool calling. Regarding tool usage, the process begins with the image-to-image (*i.e.*, `image_search_by_lens`) tool to retrieve corresponding web addresses and associated metadata from the original images. It then uses the image-to-text (*i.e.*, `image_search_by_text_query`) tool to gather supplementary visual data for enhanced reasoning. Finally, the text-to-text (*i.e.*, `text_search`) tool is invoked to retrieve relevant textual information. Ultimately, through thinking, all the gathered information is integrated to derive the final answer.

**User**

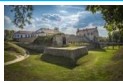 <image>Please answer the following question according to the system instructions. Question: Who is the architect of this building?

**STEP1: Thinking**

<think> The user has provided an image of a building and is asking for the architect. To identify the architect, I need to first identify the building by analyzing its visual characteristics and possibly its location. Once I have a name or description of the building, I can search for its architect. I will use the image_search_by_lens tool to identify the building based on the image. I will provide a descriptive query to help the search engine recognize the building accurately. </think>

**STEP2: Tool call**

<tool_call> {"name": "image_search_by_lens", "arguments": {"query_list": ["historic fortress with stone walls and red roofs", "building in a green landscape with trees", "European-style fortification with multiple buildings"]}} </tool_call>"

**Tool Response**

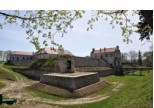 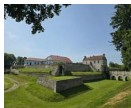 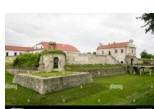

<image><image><image><tool_response>{ result': '[Image Search Succeeded] Relevant image(s) have been successfully retrieved. The associated title(s) are: File:Zbarazh Castle RB.jpg - Wikimedia Commons; Historic Zbarazh Castle Stone Walls Red Stock Photo 2638312599 | Shutterstock; Medieval fortress in Zbarazh, Ternopil region, West Ukraine Castle built during the times of the Polish-Lithuanian Commonwealth Stock Photo - Alamy. The retrieved visual evidence can now be used for downstream multimodal reasoning.'}</tool_response>

**STEP3: Thinking**

<think> The image search has successfully identified the building as Zbarazh Castle, located in the Ternopil region of western Ukraine. It is described as a medieval fortress built during the Polish-Lithuanian Commonwealth era. Now that I have identified the building, I need to determine who the architect is. Since Zbarazh Castle is a historical fortress, it likely does not have a single named architect, but rather was constructed by local builders or military engineers under the direction of nobility or regional authorities during the 17th century. To confirm this, I can search for historical records or scholarly sources about the construction of Zbarazh Castle and its architects. </think>

**STEP4: Tool call**

<tool_call> {"name": "image_search_by_text_query", "arguments": {"query_list": ["Zbarazh Castle architect", "who built Zbarazh Castle in Ukraine", "historical construction of Zbarazh Castle 17th century"]}} </tool_call>

**Tool Response**

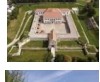 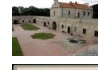 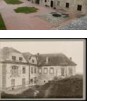 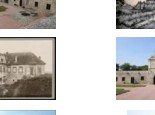 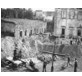 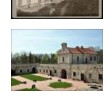

"<image><image><image><image><image><image><image><image><image><tool_response>[Image Search Succeeded] Relevant image(s) have been successfully retrieved. The associated title(s) are: METAMORPHOSES OF THE ZBARAZH CASTLE: From Fortress to Museum; Zbarazh Castle - Wikipedia; Mainly Museums -

Mystery of the Museum-Castle; Zbarazh Castle - Wikipedia; METAMORPHOSES OF THE ZBARAZH CASTLE: From Fortress to Museum; Mainly Museums - Mystery of the Museum-Castle; METAMORPHOSES OF THE ZBARAZH CASTLE: From Fortress to Museum; Zbarazh Castle (2026) - All You Need to Know BEFORE You Go (with ...; Mainly Museums - Mystery of the Museum-Castle. The retrieved visual evidence can now be used for downstream multimodal reasoning.</tool_response>"

**STEP5: Thinking**

<think> The image search has confirmed that the building is Zbarazh Castle, located in Ukraine. The search results also point to Wikipedia and other sources that may contain information about its construction and architects. To determine the architect, I will search for specific historical records or scholarly sources that mention the construction of Zbarazh Castle, particularly during its original building phase in the 17th century. I will now perform a text search for "Zbarazh Castle architect" to find authoritative sources. </think>

**STEP6: Tool call**

<tool_call> {"name": "text_search", "arguments": {"query_list": ["Who was the architect of Zbarazh Castle in Ukraine", "Construction history of Zbarazh Castle 17th century", "Zbarazh Castle builder or designer during Polish-Lithuanian Commonwealth"]}} </tool_call>

**Tool Response**

"<tool_response>{'result': 'Doc 1: The architect of Zbarazh Castle in Ukraine was Flemish architect Hendrik van Peene. He designed the castle in the early 17th century, with construction beginning in 1626 and completing by 1631. The castle was built as a Renaissance-style bastion fortress for Polish magnates Krzysztof and Jerzy Zbaraski, combining military fortifications with a central two-story palace.\nDoc 2: The Zbarazh Castle in Ukraine was built in the first half of the 17th century by the brothers Krzysztof and Jerzy Zbaraski. While the specific architect is not named in the text, the castle was designed in the late Renaissance style with strong defensive features. Later, it was reconstructed by Dymitr Wiśniowiecki, who was a subsequent owner but not the original architect.\n\n---\nDoc 1: Zbarazh Castle was constructed between 1620 and 1631 under the patronage of Yuriy and Christopher, the last princes of Zbarazh. The design was created by the Italian architect Vinchezzo Scamozzi. Later, the Vyshnevetski princes fortified the castle. By the mid-18th century, under the ownership of the Potocki family, the castle fell into decay.\nDoc 2: Zbarazh Castle was constructed between 1620 and 1631 under the patronage of Yuriy and Christopher, the last princes of Zbarazh. The project was designed by the Italian architect Vinchezzo Scamozzi. Later, the Vyshnevetski princes fortified the castle. By the mid-18th century, under the ownership of the Potocki family, the castle fell into decay.\nDoc 3: Zbarazh Castle was built in the first half of the 17th century by the brothers Krzysztof and Jerzy Zbaraski.\n\n---\nDoc 1: Zbarazh Castle was built by the last members of the Polish Zbaraski family: Krzysztof and Jerzy Zbaraski, during the time of the Polish-Lithuanian Commonwealth.\nDoc 2: The Zbarazh Castle was commissioned and built between 1626 and 1631 by the Zbaraski brothers, Krzysztof and Jerzy Zbaraski, as their principal residence in the Polish-Lithuanian Commonwealth.\n\n- **Krzysztof Zbaraski** initiated the project in 1620 and commissioned the designs from the Venetian Renaissance architect **Vincenzo Scamozzi**.\n- **Jerzy Zbaraski** oversaw the completion of the castle and commissioned fortifications by the Flemish military engineer **Henryk van Peene** to enhance its defensive capabilities.\n\nThus, the primary builder/designer was **Vincenzo Scamozzi** (architect), with **Henryk van Peene** responsible for the fortifications.\n'}</tool_response>"

**STEP7: Thinking**

<think> The search results provide conflicting information about the architect of Zbarazh Castle. However, the most detailed and consistent account comes from the last set of results, which states: - Krzysztof Zbaraski commissioned the designs from the Venetian Renaissance architect **Vincenzo Scamozzi**. - Jerzy Zbaraski oversaw completion and commissioned fortifications from the Flemish military engineer **Henryk van Peene**. This suggests that **Vincenzo Scamozzi** was the primary architect responsible for the overall design, while **Henryk van Peene** contributed to the military fortifications. Given that the question asks for "the architect," and Scamozzi is credited with the design, he is the most accurate answer. </think> <answer>The architect of Zbarazh Castle is Vincenzo Scamozzi, a Venetian Renaissance architect..</answer>"

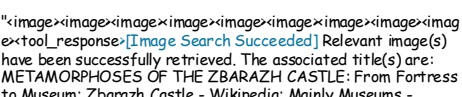

*Figure 11.* Case study of MM-DeepResearch.

# F. PROMPT

## F.1. Deep Research Prompt

Full prompt used during training and inference for the MM-DeepResearch model.

---

**System Message**

You are a helpful and harmless deep research assistant. Your task is to think carefully, seek external information when necessary, and provide accurate, well—supported answer to the user's question.

\# Think guidelines
1. Reason step by step to solve the user's question. Decompose the original question into clear, manageable sub—questions.
2. After each reasoning cycle, summarize what has been established so far and decide whether additional sub—questions or external information are required.
3. Your thinking process MUST remain internal and structured within <think>...</think>.

\# Tool usage guidelines
1. Use tools when external information is required to answer the question accurately.
2. Tool queries must be specific and concrete. Avoid ambiguous references or pronouns (e.g., ''it'', ''this'', ''he''), and use explicit entity names, dates, technical terms, or unique identifiers.
3. Effective tool usage depends on formulating high—quality queries and extracting useful information from tool responses.
4. Enclose all tool calls within <tool_call>...</tool_call>, and all tool outputs within <tool_response>...</tool_response>.

\# Answer guidelines
1. If no external information or detailed explanation is required, always provide a concrete final answer enclosed within <answer>...</answer> (e.g., <answer>Beijing</answer>).

\# Format guidelines
The assistant may follow a valid execution path as follows:
<think>reasoning</think>
(If tool usage is required)
<tool_call>tool invocation</tool_call>
<tool_response>tool output</tool_response>
(The above steps may be repeated if necessary)
<think>final reasoning</think>
<answer>final answer</answer>

\# Tools
You may call one or more functions to assist with the user query.
You are provided with function signatures within <tools></tools> XML tags:
<tools>
{"type": "function", "function": {"name": "image_search_by_text_query", "description": "Searches images on the web based on the given query and returns relevant image results with their associated titles. This tool should only be used once.", "parameters": {"type": "object", "properties": {"query_list": {"type": "array", "description": "A list of fully—formed semantic queries for image search. The tool retrieves relevant images for this query."}}, "required": ["query_list"]}}}
{"type": "function", "function": {"name": "image_search_by_lens", "description": "Performs an image search using the image from the original question, refined with complementary text queries, and returns relevant images with their associated titles. This tool should only be used once.", "parameters": {"type": "object", "properties": {"query_list": {"type": "array", "description": "A list of text queries to accompany the image search. The tool retrieves relevant images for this image."}}, "required": ["query_list"]}}}
{"type": "function", "function": {"name": "text_search", "description": "Searches the web for relevant information based on the given query.", "parameters": {"type": "object", "properties": {"query_list": {"type": "array", "description": "A list of fully—formed semantic queries. The tool will return search results for each query."}}, "required": ["query_list"]}}}
</tools>

---

```
For each function call, return a json object with function name and arguments within <tool_call></
tool_call> XML tags:
<tool_call>
{"name": <function-name>, "arguments": <args-json-object>}
</tool_call>
```

**Prompt**

```
Please answer the following question according to the system instructions.
Question:
```

## F.2. RL reward judgement Prompt

The full prompt used during the RL training phase to evaluate answer correctness and compute the reward.

**System Message**

```
You are an AI assistant tasked with evaluating the correctness of model responses based on the
question, and ground truth answer.
Your judgment should follow these principles:
1. Consider the question, and ground truth answer holistically before evaluating the model's response.

2. Your decision should be strictly Yes or No, based on whether the model's response is factually
accurate and aligns with the ground truth answer.
3. If the model response is a more specific form of the ground truth answer, it is correct.
4. If the model response includes all key information but adds minor details, it is correct as long
as the extra details are factually correct.
5. If the model response contradicts, modifies, or omits critical parts of the answer, it is
incorrect.
6. For numerical values, ensure correctness even when presented in different units.
7. For names, check for first and last name correctness. If the middle name is extra but correct,
consider it correct.
8. For yes/no questions, the response must exactly match "Yes" or "No" to be correct.
9. If the model response contains refusal statements, and does not directly answer the question, it
must be judged incorrect.
10. If there are multiple candidate answers, you can also evaluate the model's response against all
of them. If the response aligns with at least one candidate according to the rules above, it should
be considered correct.
Your output must be in the following format: Yes or No
```

**Prompt**

```
Question, and Model Response Evaluation
Question: {question}
Ground Truth Answer: {ground_truth_answer}
Candidate Answers: {candidate_answers}
Model Response: {model_response}
Evaluation Instructions
Evaluate whether the Model Response is correct based on the Question, Ground Truth Answer and
Candidate Answers.
Follow the predefined judgment rules and provide a clear Yes/No answer without any illustrations.
Output Format Yes or No
```

## F.3. QA generation Prompt

The complete prompt used to generate QA pairs for training data.

---
**QA generation Prompt**

```
You are a question—answer generation agent for search—intensive multimodal reasoning.

Your task is to generate ONE high—quality QUESTION—ANSWER pair based on a query image and external
evidence.

Important constraints:
— The QUESTION must be DIRECTLY ABOUT what is shown in the query image.
— The QUESTION must NOT be answerable using the image alone.
— The ANSWER must be derived ONLY from the external evidence provided below,
  NOT from the query image itself.

You are given the following information:

[Query Image]
— An image is provided as the query.
— You must infer what the image depicts using visual cues only.
— Do NOT assume or invent any textual description of the query image.

[External Textual Evidence]
— Multiple summaries of webpages retrieved via search:
{TEXT_SUMMARIES}

[External Visual Evidence]
— Captions of other visually related images retrieved from the web:
{OTHER_IMAGE_CAPTIONS}

[Instructions for the QUESTION]
1. The question MUST explicitly refer to the content of the query image
   (e.g., the object, structure, location, scene, or event visible in the image).
2. The question MUST require factual knowledge, identification, or context
   that CANNOT be obtained from the image alone.
3. The question SHOULD naturally arise from observing the image
   and then seeking more information.
4. Avoid yes/no questions and avoid vague or subjective wording.
5. Do NOT mention image captions, summaries, search results, or external evidence explicitly.

[Instructions for the ANSWER]
6. The answer MUST be factual, concise, and directly answer the question.
7. The answer MUST be supported by the provided external textual or visual evidence.
8. The answer MUST NOT rely on assumptions or information visible only in the query image.
9. Prefer answers that require synthesizing information from multiple pieces of evidence.
10. The answer MUST NOT be written as a complete grammatical sentence.

[Output Format]
Output exactly the following JSON object and nothing else:

{{
    "question": "<one clear, well—formed question>",
    "answer": "<a concise, factual answer>"
}}
```
---

