# OpenReview forum: "MM-DeepResearch: A Simple and Effective Multimodal Agentic Search Baseline"
_ICML.cc/2026/Conference — ICML 2026 regular_

### Official Review · Reviewer_acv9 · 2026-02-27

**Soundness:** 2
**Presentation:** 3
**Significance:** 3
**Originality:** 2
**Overall Recommendation:** 4
**Confidence:** 4

**Summary:**

The paper introduces a multimodal agentic search baseline designed to conduct multimodal deep research tasks. The authors identify three main bottlenecks: a scarcity of search-intensive multimodal QA data, a lack of effective search trajectories, and the prohibitive costs associated with training using online search APIs. To address these, the authors introduce three core components: Hyper-Search, DR-TTS and Offline Search Engine based tools.

**Compliance With Llm Reviewing Policy:**

Affirmed.

**Final Justification:**

The paper has clear motivation. The rebuttal partially solves my concern. I increase my score from 3 to 4.

**Key Questions For Authors:**

1. The distributions of different expert models are certainly not identical, so how can we ensure continuity in the reasoning process across steps?
2. Do the trajs sampled by tree structures of different experts actually improve model performance? How large is the performance gap compared with sampling from a single large API‑based model? After mixing trajs sampled from single experts models, how large is the performance difference? Does such a strategy truly lead to performance gains?
3. Since the real search API behaves differently from an offline search engine (for example, in how queries are rewritten), how can we ensure that a model trained on an offline engine can be transferred to the real engine while maintaining consistent performance improvements? Please provide a more detailed analysis.

**Limitations:**

yes

**Strengths And Weaknesses:**

Strength:
1. Hyper-Search utilizing a hyper-graph to model cross-modal relationships is a innovative and insightful method for generating multi-hop, multi-tool QA data.
2. MM-DeepResearch demonstrates significant gains over strong baselines.

Weakness:
1. Potential distribution shift: The offline search engine relies on a fixed corpus and Wikipedia, which prevents the agent from learning to handle truly real-time, zero-day information during reinforcement learning.
2. Computational Overhead of DR-TTS and Unclear Performance Gains: While the paper claims that DR‑TTS improves trajectory quality, current large API‑based models are already capable of effectively using only four tools, making the cost of training separate expert models relatively high and potentially unnecessary.
3. The ablation study is insufficient: Table 4 does not demonstrate the performance gap when each tool is used independently, and the individual performance of each expert model remains unclear.

---

> ### Author Rebuttal · Authors · 2026-03-31
>
> Dear Reviewer acv9,
>
> Thank you for your careful reading and feedback. We appreciate your recognition of our data construction and performance of MM-DeepResearch. We have carefully considered your questions and will revise paper to clarify online-offline search gap, effectiveness of DR-TTS, additional ablations, and continuity across experts.
>
> ---
> **W1&Q3. Distribution shift between offline and online search**
>
> Thank you. We agree that an offline search engine cannot fully cover truly real-time or zero-day information. However, we believe this limitation is limited. Most training and evaluation questions are constructed from existing sources, so the required evidence is usually stable rather than real-time. In RL, what matters most is that the retrieval environment can provide the evidence needed for reasoning and reward.
> > how can we ensure that a model trained on an offline engine can be transferred to the real engine while maintaining consistent performance improvements?
>
> We also had this concern initially, but empirically the distribution gap appears limited. **First**, the offline retrieval pool is partly built from real search API results, making training closer to deployment. **Second**,  the base model already has strong generalization ability, and our training mainly focuses on teaching the model how to use tools and how to integrate long-context information to reason toward the correct answer. **Finally**, although training uses the offline engine, evaluation is conducted with online search APIs, where MM-DeepResearch shows strong gains across benchmarks over prior methods trained with online search engines. This suggests that the model learns transferable search behaviors rather than overfitting to the offline engine.
>
> ---
> **W2&Q2. Effectiveness of DR-TTS**
> > current large API‑based models are already capable of effectively using only four tools
>
> Thank you for your insightful questions. We understand the intuition that existing API-based large models may seem like a simpler option for trajectory generation. However, we develop DR-TTS to address several key limitations of commercial APIs that make them less suitable for producing high-quality training trajectories.
> - Black-box distillation barrier: This is the most critical factor. Although advanced API models can use search tools, they usually expose only the final answer while hiding intermediate CoT and tool-call steps to protect proprietary capabilities. Without these fine-grained trajectories, their value for training open-source agentic models is limited.
> - Prohibitive cost: Agentic search inherently requires long context reasoning and multiple search iterations. Sampling thousands of multi-turn trajectories from commercial APIs is prohibitively expensive.
>
> > How large is the performance gap compared with sampling from a single large API‑based model?
>
> As noted above, API-based models such as GPT-5 using web search typically provide only final answers, without intermediate reasoning or tool-use steps. Training on such answer-only outputs performs worse on complex tasks than DR-TTS, which provides transparent step-by-step trajectories for learning search and reasoning behaviors.
> ||API|DR-TTS|
> |-|-|-|
> |MMSearch|31.7|52.3|
> > Do the trajs sampled by tree structures of different experts actually improve model performance?
>
> The gains mainly come from: (1) it produces transparent step-by-step CoT trajectories, including both reasoning and tool-use steps, which are much more useful for training than answer-only supervision; (2) it reduces tool-use bias by decomposing the model into single-tool experts and recomposing them through tree search, leading to more balanced tool utilization; and (3) the tree structure enables more efficient exploration of the trajectory space, helping find shorter and more effective reasoning paths.
>
> ---
> **W3. More ablation study**
>
> Thank you for this suggestion. We now report the performance of each tool when used independently in the table below. Among individual tools, text-to-text information retrieval provides the largest gain, followed by image-to-image retrieval. More importantly, combining all four tools yields the best overall performance, showing that they are complementary in our framework.
> ||w/o Tool|T2T(info)|T2I|I2I|T2T(model)|4 Tools|
> |-|-|-|-|-|-|--|
> |Acc|11.7|55.9|26.8|38.1|32.7|67.8|
>
> ---
> **Q1. Continuity across experts**
>
> Continuity is maintained through autoregressive generation: during recomposition, each expert continues from the shared dialogue context rather than treating previous steps as isolated prompts. This naturally anchors generation to the existing reasoning chain. More algorithmic details are provided in our response to **Reviewer GkdK'W2**.
>
> ---
> Lastly, thank you very much for your constructive feedback. We will incorporate these points into the revised paper. If you have any further questions, please feel free to let us know.

---

> > ### Author Rebuttal · Reviewer_acv9 · 2026-04-01
> >
> > Thank you for your rebuttal, which resolved my concern. But the alignment between local search engines and online engines still confuses me. overall, I will increase my score to support this work.

---

> > > ### Author Response · Authors · 2026-04-04
> > >
> > > Thank you very much for the follow-up. We agree that the alignment between the local/offline search engine and the real online engine should be clarified more explicitly. We also agree that the two settings are inherently different.
> > >
> > > In principle, online search is more powerful because it has access to fresher and broader information. This paper, however, is not to demonstrate that offline search is better than online search, but to show that an offline search engine can reach similar results while providing a much better trade-off between performance and search cost, avoiding the high expense of large-scale real API calls during RL training.
> > >
> > > Indeed, for the same model-generated query, offline and online engines may return different results in ranking, presentation, and even retrieved documents. However, the key requirement is not exact engine-level matching, but whether the offline environment preserves the task-relevant information needed for downstream reasoning. To reduce this gap, our offline search engine stores key webpage content collected from online sources, so that the critical information required to solve the task remains available offline.
> > >
> > > In addition, as described in Line 263 (right), the tool response content is masked during training. Therefore, the model does not directly learn engine-specific response patterns, such as ranking, formatting, or wording. Instead, what matters for RL is whether the retrieved context contains the correct evidence that the model can later integrate and reason over.
> > >
> > > From this perspective, what the model primarily learns are high-level, transferable capabilities: generating useful queries, retrieving relevant information, and synthesizing that information into the final answer. These are exactly the abilities that support transfer across different search environments.
> > >
> > >  We will revise the paper to make this point clearer. Thank you again for your helpful feedback!

---

### Official Review · Reviewer_zGk6 · 2026-03-08

**Soundness:** 2
**Presentation:** 2
**Significance:** 3
**Originality:** 3
**Overall Recommendation:** 3
**Confidence:** 4

**Summary:**

This paper proposes MM-DeepResearch, a multimodal deep research agent trained through three main ingredients: Hyper-Search for generating search-intensive multimodal QA data, DR-TTS for synthesizing search trajectories by decomposing search by tool type and recomposing tool experts through tree search, and an offline multimodal search engine to support reinforcement learning without expensive online APIs. The final system is trained with SFT plus multi-turn RL and evaluated on six benchmarks. The paper reports strong benchmark performance, especially for the 8B and 32B models, together with ablations showing gains from Hyper-Search, DR-TTS, and the combination of information-based and knowledge-based search tools.

**Compliance With Llm Reviewing Policy:**

Affirmed.

**Key Questions For Authors:**

1. Please provide a much clearer contamination analysis for the offline search engine and generated training data. Since GPT is used to generate candidate search queries and external webpages/images are pre-fetched into an offline corpus, how do you ensure that benchmark-specific answers or near-duplicate evaluation evidence are not effectively cached during training?

2. What is the actual contribution of the hypergraph structure in Hyper-Search? Please add stronger ablations that keep the same external evidence budget but remove the hyperedge formulation, so the reader can tell whether the gain comes from hypergraph structure or from more general multimodal retrieve-expand-generate-filter design.

3. Please make DR-TTS more precise algorithmically. How are search-related tasks categorized by tool type, how are the tool experts trained, what are the search depth and branching limits, what pruning rules are used, and what are the measured gains in trajectory success and diversity before downstream SFT?

4. At evaluation time, you use an auxiliary LLM to verify and summarize tool responses before feeding them back to the model. How much of the final benchmark performance depends on this helper? Please report ablations without it.

5. The knowledge-based search tool returns language-model-generated information rather than grounded retrieved evidence. How do you view this tool relative to retrieval, and how should readers interpret comparisons against systems that rely only on grounded search tools?

6. Table 7 shows that online search still clearly outperforms offline search at evaluation. What are the main failure modes of the offline engine, and what aspects of search behavior learned offline transfer well versus poorly?

7. The title calls this a simple baseline, but the full recipe is fairly elaborate and compute-heavy. What is the minimal version of the system that still performs competitively, and which components are truly essential?

**Limitations:**

No.

The paper does not sufficiently discuss its limitations. The impact statement is too brief and does not meaningfully engage with the main concerns raised by the work. In particular, the paper should discuss:
- possible contamination or leakage through offline corpus construction,
- reliance on multiple strong external models for query generation, filtering, reward judgment, and knowledge-based search,
- the gap between offline and online search performance,
- the fact that the method is not really simple despite the title,
- limited reproducibility due to under-specified trajectory synthesis details,
- compute and infrastructure cost, including many H100 GPUs and auxiliary judge/search systems.

**Strengths And Weaknesses:**

Soundness:
The paper tackles a timely and important problem. It is reasonable to focus on the lack of multimodal search-intensive data, the difficulty of obtaining good multi-tool trajectories, and the high cost of online search during RL. The overall decomposition of the solution into data generation, trajectory synthesis, and offline search infrastructure is sensible. I also appreciate that the experiments include both models without native tool-use capabilities and models that already have some agentic ability.

The paper's empirical results are encouraging. The main table shows consistent improvements over the chosen open baselines, and the ablations suggest that Hyper-Search and DR-TTS both matter. The cost comparison between online and offline search is also practically relevant.

My main concern is that several of the paper's central claims are still not supported with enough rigor or enough clean evidence. First, Hyper-Search is described as a hypergraph-based framework, but much of the actual pipeline relies on strong external models for captioning, summarization, question generation, and filtering. The current evidence does not clearly isolate what the hypergraph structure itself contributes beyond a broader retrieve-expand-generate-filter pipeline. The graph-based comparison helps, but it is still limited.

Second, DR-TTS is interesting but under-specified. The method says tasks are categorized by required tools using GPT, then tool-specific experts are trained, and finally a tree search recomposes them. But important algorithmic details are still unclear: how the categories are defined in practice, how balanced the subsets are, what search depth and branching budget are used, how pruning works, and how much improvement in trajectory success rate or diversity is obtained before downstream fine-tuning.

Third, the offline search setup raises fairness and contamination concerns. The offline corpus is built by prompting GPT to generate candidate text and image queries, then using SerpAPI, Jina, and Wikipedia to pre-fetch relevant content. This is a reasonable engineering strategy, but it makes it important to explain exactly how evaluation leakage is prevented. I did not see a sufficiently strong contamination analysis.

Fourth, the evaluation protocol itself contains an additional helper at test time: when using online search APIs, the system employs an auxiliary LLM to verify and summarize tool responses before feeding them back into reasoning. This is an important part of the final system, but I did not see a clear ablation isolating how much this helper contributes to the reported results.

Finally, the knowledge-based search tool is not grounded retrieval in the usual sense. It queries language models and returns model-generated knowledge. This may be useful in practice, but it also means some of the reported performance may come from prior model knowledge rather than search and retrieval alone. The paper should discuss this much more explicitly.

Presentation:
The paper is fairly easy to follow at a high level. The motivation is clear, and the three-part structure of Hyper-Search, DR-TTS, and offline RL is intuitive. The figures are helpful, especially the workflow figures and the case study.

However, the presentation still needs a lot of polishing. Many claims are stated strongly, but the evidence is not always as clean as the wording suggests. The paper repeatedly frames itself as a "simple and effective baseline," yet the actual recipe is fairly complex, involving synthetic data generation, specialized tool experts, tree search, offline corpus construction, SFT, RL, and an auxiliary LLM summarizer at evaluation time. This mismatch between title and method makes the contribution less crisp.

There are also many places where more concrete detail is needed for reproducibility, especially around the exact search construction pipeline, DR-TTS exploration rules, and reward-judging setup. The writing quality is acceptable but still contains noticeable grammatical issues and some imprecise phrasing.

Significance:
The problem is significant. Open multimodal research agents with search and tool use are an important direction, and reducing training dependence on expensive online APIs would be valuable for the community. The paper also touches a real bottleneck in practice: how to create search-intensive multimodal data and trajectories at scale.

That said, I see the significance as moderate rather than high in the current form. The reported gains are real, but not yet compelling enough to establish this as a strong new baseline for the field. Some baselines have missing entries, some results are self-evaluated, and the strongest system-level claims depend on multiple interacting components that are not fully disentangled. The gap between offline and online search at evaluation also suggests that the core training infrastructure still has meaningful limitations.

Originality:
The paper is reasonably original at the system level. The combination of hypergraph-style QA generation, decomposed-recomposed tool experts, and offline search for RL is not trivial. I do think there is a real idea here.

Still, the novelty of each individual component appears more moderate than the paper sometimes implies. Hyper-Search builds on familiar web expansion and synthetic data generation ideas; DR-TTS resembles a divide-and-recompose expert strategy; and the offline search engine is mainly an engineering adaptation. So I view the paper as moderately original overall, with novelty coming from the full recipe rather than from one especially sharp technical contribution.

---

> ### Author Rebuttal · Authors · 2026-03-31
>
> Dear Reviewer zGk6,
>
> Thank you for your careful reading and constructive feedback. We appreciate your recognition of the importance and practical value of our work, as well as the encouraging empirical results. We also thank you for your thoughtful comments, which we address point by point below.
>
> ---
> **W1. Contamination in search engine**
>
> Our offline search engine is designed as an information retrieval backend rather than an answer cache. We strictly separate training and testing questions, and the candidate queries used to build the offline corpus are generated only from training data rather than benchmark test questions. Thus, during training the model only interacts with information retrieved for training-related queries, not benchmark-specific evaluation queries or gold answers. For near-duplicate evidence, we deduplicate webpages with the same URL in offline search engine.
>
>
> ---
> **W2. Contribution of the hypergraph structure in Hyper-Search**
>
> We would like to clarify that we have included a comparison between graph-based construction and Hyper-Search in Tab 2. Hyper-Search improves MMSearch and increases average tool calls. This suggests that the hypergraph better captures higher-order cross-modal and cross-source relations, leading to more search-intensive QA generation.
> ||w/o hypergraph|w/ hypergraph|
> |-|-|-|
> |Avg. Tool Calls|1.7|2.3|
> |MM-Search|64.8|67.8|
>
> ---
> **W3. DR-TTS details**
> > How are search-related tasks categorized by tool type, how are the tool experts trained
>
> Search-related tasks are first categorized by prompting GPT to infer the required tool type, and each subset is then used to train a corresponding tool expert via RL without cold start.
> > what are the search depth and branching limits
>
> During recomposition, we use depth 5 and branching factor 4, with one child per tool.
> > what pruning rules are used
>
> For pruning, if a model produces an incorrect final answer, that branch is terminated. More algorithmic details are provided in our response to **Reviewer GkdK'W2**.
> > what are the measured gains in trajectory success and diversity?
>
> Compared with single-model trajectory generation, DR-TTS improves both trajectory success rate and tool-use diversity, indicating broader and more effective trajectory exploration.
> ||Sucess Rate|Diversity|
> |-|-|-|
> |Single-model|48|1.4|
> |DR-TTS|71|2.5|
>
> ---
> **W4. Analysis of auxiliary LLM at evaluation**
>
> We use an auxiliary LLM to summarize long search outputs before feeding them back to the model, since a single webpage may exceed the context limit and disrupt reasoning. The results below are without this module.
> ||MMSearch|
> |-|-|
> |w/ LLM|61.4|
> |w/o LLM|69.0|
>
> ---
> **W5. Knowledge-based tool**
>
> We introduce this tool for cases where search engines may fail to return useful evidence, especially for queries requiring domain knowledge not easily available on open webpages. For example, some specialized concepts or technical distinctions may mainly appear in textbooks or other non-web sources, while a strong language model can still answer from its parametric knowledge. We therefore view it as a complementary knowledge augmentation module and will clarify this distinction in the revision.
>
> ---
> **W6. Online vs. offline search gap at evaluation**
>
> Online search performs better at evaluation because it has access to broader, fresher, and more complete web information, while the offline engine is limited by corpus coverage. At the same time, Tab. 7 shows that search behaviors learned with the offline engine transfer effectively to online evaluation. Its main limitation is that it may fail to return useful evidence even for correct queries, reducing the optimization signal. However, this effect is limited as long as the offline engine contains sufficient task-relevant knowledge, as shown by the small gap between training with online and offline search below.
> ||Train online|Train offline|
> |-|-|-|
> |MMSearch|68.1|67.8|
>
> ---
> **W7. Simple baseline claim**
>
> - **Positioning.** By simple, we mean relatively simple and cost-effective for existing deep research agents, whose training typically requires not only substantial compute but also expensive search APIs.
>
> - **Minimal setup.** Our method balances training resources and search cost, and we view it as a relatively minimal practical setup. For the 8B version, training can be run with 6 GPUs in total: 4 for training and 2 for offline search engine and judge model.
>
> - **Key components.** In our pipeline, all three components are essential: Hyper-Search provides QA data for RL, DR-TTS provides cold-start search trajectories, and offline search engine enables cost-efficient reward-driven training.
>
> ---
> **Limitation**
>
> We will expand the impact statement accordingly and incorporate the above discussion there.
>
> ---
> Again, we sincerely thank you for these helpful comments.  We believe they will improve the paper, and we will incorporate them carefully in the revision.

---

### Official Review · Reviewer_GkdK · 2026-03-10

**Soundness:** 3
**Presentation:** 2
**Significance:** 2
**Originality:** 3
**Overall Recommendation:** 3
**Confidence:** 4

**Summary:**

This paper proposes MM-DeepResearch, a multimodal deep-research agent with three main contributions: (i) Hyper-Search, a hypergraph-based data synthesis pipeline to generate search-intensive multimodal QA, (ii) DR-TTS, a Decompose–Recompose Tool Tree Search procedure that trains tool-specific experts and recomposes them for tree-search trajectory discovery, and (iii) an offline multimodal search engine enabling cost-effective multi-turn RL (GRPO) without online APIs. The resulting models (7B/8B/32B) show strong performance across six benchmarks, with ablations indicating benefits from Hyper-Search data, DR-TTS trajectories, and tool composition, as well as significant training cost/latency reductions from the offline search engine.

**Compliance With Llm Reviewing Policy:**

Affirmed.

**Key Questions For Authors:**

see weakness

**Limitations:**

see weakness

**Strengths And Weaknesses:**

##Strength:

-Systematic Data Construction: The paper provides a comprehensive framework for generating training data. Hyper-Search effectively models the relationships between visual and textual nodes via hyperedges, ensuring that the generated QA pairs require multi-step, multi-modal evidence gathering. DR-TTS effectively generates complex search trajectories by decomposing tool-use tasks and recomposing tool experts via tree search

-Extensive and Reasonable Experimental Validation: The authors evaluate their model across a wide range of benchmarks (SimpleVQA, MMSearch, LiveVQA, FVQA, InfoSeek, and Browsecomp-VL). Baselines include both closed-source models (GPT-4o/5) and state-of-the-art open-source agentic search models (WebWatcher, SenseNova-MARS).

##Weakness:

-Limited Architectural Novelty: The overall framework of agentic search (thinking -> search -> observation -> final answer) has been explored by recent works such as Search-R1 and MM-Search-R1. Furthermore, using an offline retrieval environment for RL training is increasingly becoming a standard practice in the community to ensure stability and efficiency.

-Lack of Technical Detail: The "Decompose-Recompose" algorithm lacks a formal definition or detailed pseudo-code. It is unclear how the "experts" are specifically specialized beyond being trained on different subsets. Moreover, the "recomposition" phase—where these experts collaboratively explore and merge their outputs into valid search trajectories—is described vaguely, lacking formal definitions or pseudo-code to clarify how the transitions and interactions between different tool experts are managed.

-Limited Statistical Rigor: Several evaluations (e.g., SimpleVQA, LiveVQA) are conducted on 300-sample subsets rather than the full datasets. While common in some preliminary studies, this limits the statistical power of the results and may not fully reflect the model's performance on the tail of the data distribution

---

> ### Author Rebuttal · Authors · 2026-03-31
>
> Dear Reviewer 8nc4,
>
> Thank you for your valuable comments and constructive suggestions. We appreciate your recognition of our systematic data construction framework and the extensive validation across six benchmarks. We have carefully considered your suggestions and will revise the manuscript to further clarify the novelty of our approach, add formal definitions and pseudo-code, and strengthen the evaluation accordingly.
>
> ---
> **W1. Novelty of MM-DeepResearch**
>
> Thank you for your comment. We agree that the high-level paradigm of agentic search—reasoning, search, observation, and answer synthesis—has been explored in prior work such as Search-R1 and MM-Search-R1. Our intention is not to claim novelty for this general paradigm itself, but rather for the comprehensive pipeline we have constructed specifically for multimodal deep research.
>
> Our contributions lie in three aspects: (1) **Hyper-Search**, a hypergraph-based multimodal data synthesis pipeline that organizes visual and textual evidence through higher-order relations, enabling the construction of search-intensive multimodal QA data; (2) **DR-TTS**, a Decompose–Recompose Tool Tree Search procedure that trains tool-specialized experts and recomposes them to discover effective tree-structured search trajectories; and (3) an **offline multimodal search engine** that supports scalable and cost-effective multi-turn RL training without relying on online APIs.
> We will clarify this distinction more explicitly in the revised version.
>
> ---
> **W2. Technical detail in DR-TTS**
>
> Thank you for pointing this out. We agree that formal definitions and pseudo-code would improve the clarity of DR-TTS, and we will add them in the revised manuscript. Below we clarify both the expert specialization and the recomposition phase.
> > how the "experts" are specifically specialized beyond being trained on different subsets.
>
> The specialization of each expert lies in tool-specific mastery. Instead of training a single model to handle multiple tools simultaneously, we use RL to train distinct experts, each dedicated to one specific tool. This decomposition reduces learning difficulty and improves per-tool proficiency without requiring a cold start.
> > the "recomposition" phase—where these experts collaboratively explore and merge their outputs into valid search trajectories—is described vaguely
>
> For the recomposition phase, we deploy all specialized experts concurrently using vLLM to perform collective tree search. We define the search tree as follows:
> - Root Node (S0): The initial user question.
> - Intermediate Nodes (St): A tuple consisting of `[thinking, tool_call, tool_response]`.
> - Leaf Nodes (ST): A tuple consisting of `[thinking, final_answer]`.
>
> At each step, the shared context is broadcast to all experts, and each expert proposes one next-step candidate based on its specialized tool. Thus, if there are K experts, each node expands to K candidate child nodes. The search terminates once any expert produces a final answer that is verified to be correct. The recomposition pseudocode is provided below.
>
> ```
> Input: User Question Q, Tool Experts {E1, E2, ..., EK}
> Output: A valid search trajectory
>
> 1: Initialize root node S0 = [Q]
> 2: Initialize active nodes L = [S0]
> 3: While L is not empty and max_depth not reached:
> 4:     new_nodes = []
> 5:     For each node St in L:
> 6:         For each expert Ek:
> 7:             thinking, action = Ek(St)
> 8:             If action is final_answer:
> 9:                 If VerifyCorrectness(action) == True:
> 10:                    Return trajectory(S0 -> ... -> St+1)
> 11:             Else:
> 12:                 response = Execute(action)
> 13:                 Create St+1 = [thinking, action, response]
> 14:                 new_nodes.append(St+1)
> 15:     L = new_nodes
> ```
>
> ---
> **W3. Statistical rigor and Full Dataset Evaluation**
>
> Thank you for your concern. We used the 300-sample subsets for SimpleVQA and LiveVQA to align with prior strong baselines such as WebWatcher, SenseNova-MARS, and MM-Search-R1. In particular, our subset exactly matches the samples used by WebWatcher. This protocol is common in multimodal agentic search because API-based evaluation is costly and inference is slow, and it helps ensure a fair comparison.
>
> We agree that evaluation on the full benchmarks provides stronger statistical support. To address this, we evaluated our models on the full datasets. As shown below, the full-set results remain highly consistent with the subset evaluations. We will include these results in the revised paper.
> ||SimpleVQA|MMSearch|LiveVQA|FVQA-test|InfoSeek|Browsecomp-VL|Avg|
> |-|-|-|-|-|-|-|-|
> |MM-DeepResearch-8B|65.3|67.8|65.8|69.2|72.3|37.9|63.0|
> |MM-DeepResearch-32B|67.3|69.0|68.4|70.1|73.9|43.0|65.3|
>
> ---
> Many thanks for your professional, detailed, and valuable reviews! Please let us know if you have any other questions. We will actively join the discussion until the end of the rebuttal period.

---

> > ### Author Rebuttal · Reviewer_GkdK · 2026-04-05
> >
> > Thanks for your efforts, and I will keep my score. I still have some concerns:
> >
> > Based on the interaction paradigm between large models such as Search-R1 and tools, the authors propose a Decompose–Recompose Tool Tree Search scheme, combined with a self-developed offline training paradigm. This offline training approach mitigates the problem of low training efficiency caused by tool instability to a certain extent.
> > However, it remains to be verified whether this method can effectively overcome tool instability and how the model's reflection and optimization capability performs in the online testing phase. Furthermore, it requires further investigation whether the tree search algorithm proposed by the authors leads to a reduction in inference speed.

---

> > > ### Author Response · Authors · 2026-04-08
> > >
> > > Thank you very much for your thoughtful reply and follow-up questions.
> > >
> > > ---
> > >
> > > > This offline training approach mitigates the problem of low training efficiency caused by tool instability to a certain extent. However, it remains to be verified whether this method can effectively overcome tool instability.
> > >
> > > We believe there is a misunderstanding regarding the motivation of our offline training approach. Our goal is not to address low training efficiency caused by tool instability. Rather, the main purpose is to **avoid the high cost of commercial search APIs in RL training**. In practice, the offline setup saves several hundred dollars per RL run while still achieving strong results. Moreover, training with our offline search engine outperforms prior methods trained with online search APIs.
> > > >how the model's reflection and optimization capability performs in the online testing phase
> > >
> > > We would like to clarify that this has already been reported in **Table 1**. MM-DeepResearch 8B outperforms the previous state-of-the-art SenseNova-MARS-8B across all six benchmarks by 3.4 points, while the 32B version achieves comparable performance. In addition, our 32B model surpasses WebWatcher 32B on SimpleVQA and MMSearch by 8.6 and 13.7 points, respectively. Besides, **Figure 11** further provides quantitative results under online search settings, demonstrating the model’s multi-round search and reasoning ability.
> > > > it requires further investigation whether the tree search algorithm proposed by the authors leads to a reduction in inference speed.
> > >
> > > We also clarify that the proposed Decompose-Recompose Tree Search is used only for data generation, rather than inference. Although tree search increases the time required for synthetic data construction, we consider this overhead acceptable given the resulting data quality improvements. In particular, DR-TTS yields clear performance gains over alternative approaches (please refer to our response to Reviewer acv9'W2). Importantly, **tree search is not involved during inference**, where the model performs direct reasoning only. Therefore, our method does not incur any inference-time slowdown.
> > > | Benchmark | API | DR-TTS |
> > > |---|---:|---:|
> > > | MMSearch | 31.7 | 52.3 |
> > >
> > > ---
> > >
> > > Thank you again for your helpful review.

---

### Official Review · Reviewer_8nc4 · 2026-03-16

**Soundness:** 3
**Presentation:** 3
**Significance:** 3
**Originality:** 2
**Overall Recommendation:** 4
**Confidence:** 4

**Summary:**

Starting from three key limitations of current MM DeepResearch Agents: 1. Scarcity of search-intensive multimodal QA data.  2. Lack of effective search trajectories.  3. Prohibitive cost of training with online search APIs. This paper proposes Hyper-Search, a hypergraph-based QA generation method and DR-TTS, a method that first decomposes search-involved tasks and optimizes specialized search tool experts for trajectories generation via tree search together with an offline engine supporting search tools. With the three components, the paper proposes MM-DeepResearch, a sota multimodal deep research agent, evaluated on extensive benchmarks.

**Compliance With Llm Reviewing Policy:**

Affirmed.

**Final Justification:**

The rebuttal has partially solved my problems. I will keep my score since the score is already positive.

**Key Questions For Authors:**

1. Line 145 right column, *“For text node expansion, we use an MLLM to extract the top-K informative webpage URLs from the native page content”.* Did you use MLLMs to extract **hyperlinks** from the native page and extract the content in hyperlinks as text node expansions? If so, did you evaluate the hallucinations in extracting hyperlinks? Also, the same hallucination problem for “image node expansion of Text Node”.
2. In Figure 2, what is $I_{D, 1}$? what is the definition of subscript $D$? (According to Sec. 3.1.1, the first subscript refers to the expansion depth.)
3. In Sec. 3.1.1, nodes are defined with lower cases i and t. In Figure 2, nodes are presented in upper cases I and T. Are the corresponding or not? If not, please elaborate.
4. Line082: “…to effectively synthesis of …” → …to effectively synthesize …?

**Limitations:**

See Weakness

**Strengths And Weaknesses:**

Strengths:
1. The work is well motivated. It identifies the core bottlenecks plaguing multimodal deep research with targeted methodological designs.
2. Constructs a cost-free offline multimodal search engine, enabling scalable agentic RL training without online API reliance and greatly reducing training cost and latency.

Weaknesses:
1. Six datasets are reported in main results and only MMSearch is reported in Ablation Studies. More comprehensive results are encouraged.
2. Text-to-Text tools are implemented with “domain-specific expert models”. Are hallucinations involved in the process? Specific evaluations are encouraged.
3. Some notations are unclear. (See Questions)
4. A potential missing reference and discussion to “WebSailor-V2”. The method for QA generation via knowledge graphs are similar.

---

> ### Author Rebuttal · Authors · 2026-03-31
>
> Dear Reviewer 8nc4,
>
> Thank you for your thoughtful and constructive review. We appreciate your positive comments on the motivation of our work and the value of building a scalable multimodal deep research agent without costly online search APIs. We also thank you for your detailed suggestions on experimental coverage, tool expert hallucination, notation clarity, and related work. We will revise the manuscript accordingly to improve the presentation.
>
> ---
> **W1. More comprehensive ablation results**
> Thank you. We have added more benchmarks to the ablation study below. The results show consistent conclusions across different benchmarks. Due to the response length limit, more results will be included in revision.
> ||SimpleVQA|MMSearch|LiveVQA|FVQA-test|InfoSeek|Browsecomp-VL|
> |-|-|-|-|-|-|-|
> |MM-DeepResearch-SFT|59.2|52.3|52.7|59.4|61.8|31.3|
> |MM-DeepResearch-RL|63.4|65.2|63.4|65.8|70.1|33.0|
> |MM-DeepResearch|65.9|67.8|65.0|69.2|72.3|37.9|
>
> ---
> **W2. Hallucination concerns in text-to-text tools**
>
> Thank you for raising this concern. Hallucinations may occur in text-to-text tools implemented with domain-specific expert models, just as retrieval-based search APIs may also return noisy or irrelevant evidence. We view this as a general challenge in open-domain search rather than one unique to our method. To mitigate this issue, we introduce a verification module during evaluation to check returned search content before feeding it into the reasoning process. We further evaluate its impact by removing the verification module and directly feeding the potentially hallucinated responses into the model. The results below show that the effect is relatively small, suggesting that the model can assess whether the returned information is useful from the context and then decide the next action accordingly. We will clarify this more explicitly in the revised paper.
> ||w/o T2T(Know.)|w/ T2T(Know.)+w/o verify|w/ T2T(Know.)|
> |-|-|-|-|
> |MMSearch|66.9|67.5|67.8|
>
> ---
> **W4. Reference and discussion**
>
> **Reference**: Thank you for pointing this out. We will add the citation for WebSailor-V2.
>
> **Discussion**: Compared with WebSailor-V2, Hyper-Search differs in three main aspects: (1) it is natively multimodal, jointly modeling image and text nodes for multimodal QA generation; (2) it is image-centered, which is important for multimodal deep-research tasks where the query or key evidence is visual; and (3) its nodes are grounded in URL-level web sources, making the expansion process more directly aligned with realistic multimodal web navigation and cross-source evidence aggregation. We will clarify this comparison more explicitly in the revised version.
>
> ---
> **W3. Clarify Notations**
>
> Thank you for pointing out these notation issues. We address them point by point below.
>
> **Q1: Clarification on text-node expansion, hyperlink extraction, and hallucination**
>
> Thank you for this question.
> > Did you use MLLMs to extract hyperlinks from the native page and extract the content in hyperlinks as text node expansions?
>
> As described around L 356, after obtaining URLs from SerpAPI, we use the Jina Reader API to fetch the corresponding raw webpage content; this is a standard retrieval step and does not itself introduce hallucination.
> We then use an MLLM to (1) summarize long webpage content for downstream QA generation and (2) extract URLs in the returned content, since the hyperlinks are often interleaved with surrounding text and are difficult to isolate reliably with simple rules.
> > Did you evaluate the hallucinations in extracting hyperlinks? did you evaluate the hallucinations in extracting hyperlinks?
>
> To assess hallucination in this step, we manually inspected 25 randomly sampled cases. We did not observe obvious cases of hallucinated webpage content from Jina, hallucinated URLs, or fabricated summarized webpage content. This suggests that hallucination in this stage is limited in practice. We will include these results in the revision.
> |Check Item|Result|
> |-|-|
> |Randomly sampled cases|25|
> |Hallucinated webpage content from Jina|0|
> |Hallucinated URLs|0|
> |Fabricated summarized webpage content|0|
>
> ---
> **Q2. The definition of subscript $D$**
>
> We apologize for the confusion. Here, $D$ denotes the maximum expansion depth of the hypergraph. We will clarify this notation explicitly in Sec. 3.1.1 and the caption of Fig 2.
>
> ---
> **Q3. Lowercase i,t in Sec. 3.1.1 vs uppercase I,T in Fig 2**
>
> Thanks. These symbols are corresponding and refer to the two node modalities, namely image nodes and text nodes. We will unify the notation throughout the paper to avoid confusion.
>
> ---
> **Q4. Wording issue in L 82**
>
> Thank you. We will revise **"to effectively synthesis"** to **"to effectively synthesize"**.
>
> ---
> Lastly, thank you very much for your constructive feedback and suggestions. We will incorporate these points into the revised paper. If you have any further questions, please feel free to let us know. We will be available throughout the rebuttal period.

---

### Decision · Program_Chairs · 2026-04-30

**Decision:**

Accept (regular)

**Comment:**

This paper proposes a multimodal research agent called MM-DeepResearch, that consists of three contributions: 1) a hypergraph-based QA generation method called Hyper-Search, that synthesizes multimodal QA pairs that require search tools to solve; 2) a method called Decompose-Recompose Tool Tree Search (DR-TTS) that decomposes search tasks into categories involving different tools and trains tool-specific experts and composes them to discover tree-search trajectories; and 3) an offline search engine supporting multiple search tools, that allows for RL training without using online search APIs.

The MM-DeepResearch framework achieves strong results across six benchmarks, substantially outperforming baselines. The authors also provide ablation studies over the components of the method.

Reviewer 8nc4 found that the paper is well-motivated and introduces targeted methodological designs as well as a cheap offline multimodal search engine.

Reviewer GkdK found that the paper proposes a comprehensive framework for generating training data, and that DR-TTS can effectively generate complex search trajectories. The reviewer also appreciated the breadth of the experiments, on six benchmarks and including both closed-source and open-source models as baselines.

Reviewer zGk6 found that the paper addresses an important problem and has promising empirical results with consistent improvements over the baselines.

Reviewer acv9 found Hyper-Search to be an innovative method for synthetic data generation, and found the results of MM-DeepResearch to be strong.

The reviewers also raised concerns regarding clarity, evaluations, and the gap between the online and offline search environments. However, most of these concerns were sufficiently addressed by the rebuttal.

Reviewer 8nc4 raised concerns regarding the limited empirical evaluation with only six datasets in Table 1, and only MMSearch used for ablations. The reviewer noted that the text-to-text tools are implemented using domain-specific models that could have hallucinations. The reviewer also pointed out that the paper is missing a discussion of WebSailor-V2.

The rebuttal provided ablation results on more datasets. It also acknowledged that the text-to-text tools can have hallucinations and introduced a verification module to check the content returned by the search module before feeding it into the reasoning process; the authors ablated the use of this verification module, showing little effect, which is a positive sign that the model is capable of determining whether the returned information is useful even without the verification module. The rebuttal also provided a discussion of the differences between WebSailor-V2 and Hyper-Search. Overall, the reviewer’s concerns were addressed by the rebuttal.

Reviewer GkdK raised concerns regarding the novelty of the framework and the lack of detail in the exposition. This reviewer also noted that the experiments are evaluated on 300-sample subsets rather than full datasets.

The authors’ rebuttal clarified the novelty of their pipeline, and noted that the evaluation on 300 sample subsets aligns with prior works.

Overall, the rebuttal addresses most of the concerns of Reviewer GkdK. While the reviewer did not respond to the last author rebuttal (in which the authors addressed the reviewer’s remaining follow-up questions after the first rebuttal round), and the reviewer did not enter a final recommendation, I think the rebuttal sufficiently addresses their concerns.

Reviewer zGk6 noted that the paper does not isolate what the hypergraph structure contributes; that algorithmic details about DR-TTS are missing; that the paper is missing an ablation over the auxiliary LLM that verifies tool responses before feeding them back into reasoning; and that the knowledge-based search tool is not grounded retrieval as it uses results generated by an LLM.

The authors’ rebuttal addresses most of the questions the reviewer had. The reviewer did not provide a rebuttal acknowledgement or a final justification, however I think that the rebuttal fully resolves the reviewer’s concerns.

Reviewer acv9 had concerns regarding distribution shift, the computational overhead of DR-TTS, and insufficient ablations.

The authors’ rebuttal mostly addressed the reviewer’s concerns. In addition, I think that the final response from the authors sufficiently addresses the last remaining question in the reviewer’s rebuttal acknowledgment, so the reviewer’s concerns seem to be fully resolved.

Overall, this paper presents a well-motivated and comprehensive pipeline for multimodal deep research that integrates three well-designed components (HyperSearch, DR-TTS, and an offline search engine) and achieves strong performance across several benchmarks. This paper will be of interest to the ICML community.